# WHAT TO ALIGN IN MULTIMODAL CONTRASTIVE LEARNING?

**Benoit Dufumier**[1,2*]   **Javiera Castillo Navarro**[1,3*]   **Devis Tuia**[1]   **Jean-Philippe Thiran**[1,4]

[1] EPFL   [2] NeuroSpin, CEA   [3] CEDRIC, CNAM   [4] Radiology Department, CHUV

## ABSTRACT

Humans perceive the world through multisensory integration, blending the information of different modalities to adapt their behavior. Alignment through contrastive learning offers an appealing solution for multimodal self-supervised learning. Indeed, by considering each modality as a different view of the same entity, it learns to align features of different modalities in a shared representation space. However, this approach is intrinsically limited as it only learns shared or redundant information between modalities, while multimodal interactions can arise in other ways. In this work, we introduce **CoMM**, a **Co**ntrastive **M**ulti**m**odal learning strategy that enables the **comm**unication between modalities in a single multimodal space. Instead of imposing cross- or intra- modality constraints, we propose to align multimodal representations by maximizing the mutual information between augmented versions of these multimodal features. Our theoretical analysis shows that shared, synergistic and unique terms of information naturally emerge from this formulation, allowing to estimate multimodal interactions beyond redundancy. We test CoMM both in a controlled and in a series of real-world settings: in the former, we demonstrate that CoMM effectively captures redundant, unique and synergistic information between modalities. In the latter, we show that CoMM learns complex multimodal interactions and achieves state-of-the-art results on seven multimodal tasks. Code is available here.

## 1 INTRODUCTION

Multisensory or multimodal learning (Baltrušaitis et al., 2018) involves extracting and processing information from multiple sources (*e.g.* text, audio, images, tabular data, *etc*.). The whole human experience is inherently multimodal: we simultaneously see, hear, smell, taste and feel, and these different sensory signals are combined to give us the necessary information to explore our environment. Many of the simplest tasks we tackle in our daily lives are multimodal:the way we perceive the flavor of our food or drinks does not depend solely on our taste, but also on what we see (Morrot et al., 2001) or what we hear (Woods et al., 2011) while we eat. McGurk & MacDonald (1976) have also shown that visual stimuli interact with audio signals to perform human speech recognition.

Despite the inherent multimodality of sensory systems, machine learning has largely concentrated on single-modality models, with few exceptions in areas like audio-visual speech recognition (Yuhas et al., 1989; Ngiam et al., 2011), multimedia content retrieval (Atrey et al., 2010; Snoek & Worring, 2005), and video-based human behavior analysis (Kraaij et al., 2005). Nowadays, with the emergence of self-supervised strategies and their impressive capacities for learning representations in computer vision (Chen et al., 2020a; He et al., 2020; Caron et al., 2021), NLP (Radford et al., 2018; Devlin et al., 2019) or audio (Oord et al., 2018; Niizumi et al., 2021), the paradigm has shifted to learning general multimodal representations from unlabeled data and then fine-tune the models to specific multimodal tasks. Recent works have shown success at training multimodal representations by using cross-modal contrastive objectives (Radford et al., 2021; Jia et al., 2021) to align the representations in a shared embedding space. However, this training strategy only works under the *multiview redundancy* assumption, *i.e.*, considering that all task-relevant information is shared between modalities and redundantly contained in either one of them separately. In particular, for vision-language tasks, this can be seen as a clever way to perform supervised learning on a visual encoder, which explains their success to transfer to visual classification tasks.

---

*denotes equal contribution. Contact information: {name.surname}@epfl.ch

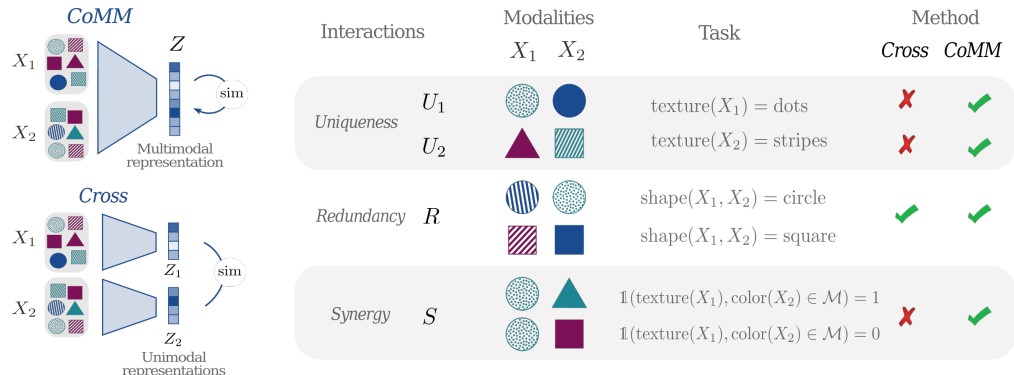

a) Multimodal models                                          b) Multimodal interactions

Figure 1: a) We propose **CoMM, a contrastive multimodal approach that allows the interplay of multiple modalities and learns multimodal interactions**. Unlike previous multimodal models (Cross) that align cross-modal features, CoMM aligns multimodal features in a shared representation space. b) Multimodal interactions are task-dependent, thus a model needs to capture all of them to generalize to any multimodal task. **CoMM's new paradigm captures multimodal interactions beyond redundancy.**

Nonetheless, these solutions are insufficient in many cases, as the interactions between modalities can arise in several ways to perform a specific task (Bertschinger et al., 2014): *redundancy* ($R$) arises when the task can be performed using either of the modalities because they contain redundant information; *uniqueness* ($U$) appears when only one of the modalities contains all the necessary information to complete the task; *synergy* ($S$) emerges when both modalities have complementary information, and they are needed simultaneously to fulfill the task. Modeling these interactions to perform multimodal learning is highly challenging as the interplay between $R, S$ and $U$ is task-dependent and difficult to measure in complex real-life scenarios. Fig. 1b shows simple tasks where the predominant type of interaction can be easily identified. We observe that we need to model specific interactions for the same input modalities to perform a specific task. Therefore, a model must capture all these terms to learn task-agnostic multimodal representations.

To achieve task-agnostic multimodal learning, we propose a **Co**ntrastive **M**ulti**M**odal self-supervised pre-training method (CoMM) that enables the **comm**unication of modalities in a single multimodal space. Unlike previous contrastive multimodal methods that impose cross-modal constraints to align unimodal representations, we propose to leverage a simple multimodal architecture to fuse multimodal inputs into a common representation and then align the multimodal features by maximizing the mutual information between augmented versions of these features (see Fig. 1a). CoMM enables to model multimodal interactions –including redundancy, uniqueness and synergy– in the context of multimodal representation learning , as these terms naturally emerge from our contrastive multimodal formulation.

CoMM's formulation is well-aligned with the global workspace theory (Baars, 1988; Goyal & Bengio, 2022) in cognitive neuroscience, which considers the nervous system as a set of multiple specialized processors working in parallel and claims the existence of a shared representation, which can be modified by any selected processor and whose content is broadcast to all processors. By analogy, CoMM considers a shared representation space built from parallel streams of modality-specific processors. When a task requires knowledge from a given modality or interactions between modalities, only these parts of the representation space should be used.

Based on Partial Information Decomposition (Williams & Beer, 2010; Bertschinger et al., 2014), we built a strong theoretical basis for CoMM to learn multimodal interactions in a self-supervised way. Empirically, we show that CoMM effectively captures redundant, unique and synergistic information between modalities in a controlled environment, where the type of interaction is known. Then, in a series of real-world datasets –from different domains (healthcare, robotics, *etc.*) and including diverse data types (image, text, audio, *etc.*)–, CoMM achieves state-of-the-art results on seven multimodal tasks with two or three modalities. In all cases, CoMM showed to be a versatile framework capable of handling any number of modalities, various data types, and different domains.

## 2 QUANTIFYING MULTIMODAL INTERACTIONS

**Problem setup.** Let $X_1, X_2, \ldots, X_n$ be random variables representing $n$ different data modalities (e.g. images, text, audio, tabular data, etc.), and a given task $Y$. Our goal is to learn a latent variable $Z = f(X)$ that is a good representation of $X = (X_1, \ldots, X_n)$ for $Y$.

For our theoretical analysis, we set $n = 2$, as multimodal interactions have not been characterized yet by PID for larger $n$. In practice, CoMM's implementation for $n > 2$ is straightforward and tested in Section 4.3. All proofs can be found in Appendix G.

For $Z$ to be a good representation of $X$ it should capture the *task-relevant* information that $X$ contains. Therefore, we need to model the information between the joint variable $X$ and the task $Y$: $I(X; Y) = I(X_1, X_2; Y)$.

**Partial information decomposition** (PID) (Williams & Beer, 2010; Bertschinger et al., 2014) states that multivariate mutual information $I(X_1, X_2; Y)$ is decomposed into three forms of interactions:

  (i) **Uniqueness.** This term appears when the task $Y$ can be completed by leveraging only one of the modalities. $U_1$ (resp. $U_2$) refers to the case when $X_1$ (resp. $X_2$) contains all task-relevant information.
 (ii) **Redundancy.** When $X_1$ and $X_2$ contain the same information about $Y$. $R$ corresponds to this redundant or shared information.
(iii) **Synergy.** Noted by $S$, this term only emerges when $X_1$ and $X_2$ are simultaneously present, because they bring different and complementary task-relevant information.

Thus, the information that $(X_1, X_2)$ has about $Y$ can be written as the contribution of four terms:
$$I(X_1, X_2; Y) = R + S + U_1 + U_2 \tag{1}$$
Moreover, Eq. (1) and the application of the chain rule of mutual information yield the following consistency equations between $R, S, U_1$ and $U_2$:
$$I(X_1; Y) = R + U_1, \qquad I(X_2; Y) = R + U_2, \qquad I(X_1; X_2; Y) = R - S \tag{2}$$

Existing methods using contrastive objectives to learn multimodal representations (Jia et al., 2021; Radford et al., 2021; Tian et al., 2020a) impose cross-modal constraints by maximizing an estimator of $I(X_1; X_2)$ as an approximation of $I(X_1, X_2; Y)$. However, this strategy is limited by the *multiview redundancy* assumption.

**Definition 1 (*Multi-view redundancy*)** $\exists \, \varepsilon > 0$ *such that* $I(Y; X_1 | X_2) < \varepsilon$ *and* $I(Y; X_2 | X_1) < \varepsilon$.

This assumption states that most task-relevant information is shared across modalities and the non-shared information is (at most) a small $\varepsilon$. In other words, any of the modalities contains enough information to fulfill the downstream task $Y$, and they can provide some kind of "supervision" to one another, which explains their success for zero-shot image classification (El Banani et al., 2023).

**Lemma 1** *Under the multiview redundancy assumption, cross-modal contrastive learning methods are limited to only learn the redundant information R.*

What happens when other sources of multimodal information intervene? FactorCL (Liang et al., 2023b) is a good initial step to integrate *uniqueness* and *redundancy* into multimodal contrastive learning by applying multimodal augmentations. However, it heavily relies on the assumption that *optimal multimodal augmentations* can be obtained by applying a two-step process based on conditional augmentations. We argue that this hypothesis is unrealistic, as the first unique augmentation is strongly related to task-relevant information of different modalities. For example, if the text caption is *"a yellow flower"*, then color jittering should not be applied to an image depicting a flower. Besides, the factorized formulation of FactorCL is impractical as it is prone to cumulative errors. Finally, this method does not consider the *synergy* between modalities.

In contrast, we propose a model that relies solely in the main hypothesis of contrastive learning, extended to the multimodal case, without relying on strong assumptions about multimodal relationships nor conditional augmentations.

**Assumption 1 (*Minimal label-preserving multimodal augmentations*)** *We assume the existence of* $\mathcal{T}^\star$, *a set of multimodal augmentations such that for any* $t \in \mathcal{T}^\star$ *and* $X' = t(X)$, *we have* $I(X, X') = I(X, Y)$.

Even if Assumption 1 might seem strong at first glance, it actually makes sense in the context of multimodal representation learning. Indeed, coming back to the example of the flower image with "a yellow flower" as a caption, applying color jittering to the image would allow the model to focus on other interactions (uniqueness or synergy) rather than color redundancy or even to refocus on other features (like the flower shape). This is discussed more in-depth in Appendix C.4.

Moreover, our assumption allows for a larger spectrum of augmentations, without being constrained to the set $\mathcal{T}_c^\star = \{t(X) = (t_1(X_1), t_2(X_2))\}$ of transformations that can be decomposed in independent unimodal augmentations.

## 3   CoMM: Contrastive Multimodal learning

We aim to learn multimodal representations that are transferable to any multimodal task. Contrastive learning has shown promising results in multimodal learning. However, current approaches fail to capture multimodal interactions other than redundancy, as shown in Section 2.

Our strategy builds upon multiview contrastive learning theory and extends it to the multimodal case. It is based on two main components:

  (i) A multimodal architecture, with specialized encoders to process any data type, and an attention-based fusion module to obtain a final **multimodal representation**.
 (ii) A **contrastive objective** that naturally captures unique, redundant, and synergistic interactions between different data modalities.

### 3.1   Towards effective multimodal representations

To obtain robust, task-agnostic and common representations $Z$ that capture uniqueness, redundancy and synergy from different input modalities, we design $f_\theta$ –a neural network parameterized by $\theta$– such that $Z_\theta = f_\theta(X) = f_\theta(X_1, X_2)$.

We define $X' = t(X)$ with $t \in \mathcal{T}$ a stochastic mapping[1] (multimodal augmentation) of $X$ and $Z'_\theta = f_\theta(X')$.

Given data processing inequalities for Markov chains $X \to X' \to Z'_\theta$ and $Z'_\theta \to X \to Z_\theta$, we have:

$$I(Z_\theta; Z'_\theta) \leq I(X, Z'_\theta) \leq I(X, X') \tag{3}$$

With these inequalities, we can prove the following lemmas:

**Lemma 2** *By optimizing $f_\theta$ to maximize $I(Z_\theta; Z'_\theta)$, and if we assume an expressive enough network $f_\theta$, we have at optimum:*

$$I(Z_{\theta^\star}, Z'_{\theta^\star}) = I(X, X') \tag{4}$$

**Lemma 3** *Let $f_{\theta^\star}$ be optimal, i.e. $f_{\theta^\star}$ maximizes $I(Z_\theta, Z'_\theta)$. Then, we have the equality $I(Z'_{\theta^\star}; Y) = I(X'; Y)$. If we consider the special case $\mathcal{T} = \{t_i\}$ such that $X' = t_i(X) = X_i$ and $Z'_{\theta^\star} = f_{\theta^\star}(X_i) = Z_i$ for $i \in \{1, 2\}$, then it follows:*

$$I(Z_i; Y) = I(X_i; Y) = R + U_i \tag{5}$$

Lemma 3 implies that optimal representations $Z_i$ preserve all the task-relevant information contained in modality $i$. Interestingly, we do not require Assumption 1 for this equality to hold.

The previous theoretical developments lead us to the key ingredients for CoMM's contrastive objectives to succeed at capturing multimodal interactions (see Section 3.3 for practical implementation):

  (i) Following Lemma 2, $U_1 + U_2 + R + S = I(X, Y)$ can be learned by optimizing the term $I(Z_\theta, Z'_\theta)$ for $\mathcal{T} = \mathcal{T}^\star$, since $I(X, X') = I(X, Y)$ by Assumption 1;
 (ii) Thanks to Lemma 3, $R + U_i$ for $i \in \{1, 2\}$ can be directly learned by optimizing the term $I(Z_\theta, Z'_\theta)$ for $\mathcal{T} = \{t_i\}$.

---

[1]Here, $\mathcal{T}$ can be any set of mappings, *e.g.*, $\mathcal{T} \neq \mathcal{T}^\star$.

## 3.2 MULTIMODAL ARCHITECTURE

Our architecture for multimodal representation learning is presented in Fig. 2. In order to capture multimodal interactions, the model consists of mainly three components:

**Encoders.** Each modality is encoded independently by one of the $n$ modality-specific encoders.

**Latent converters.** Linear modules that transform features into modality-specific sequences of embeddings. After the latent converters, a concatenation operation gathers these sequences to be fed into a transformer architecture.

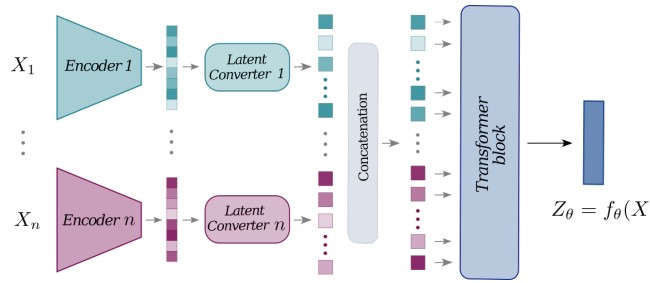

Figure 2: **CoMM's model architecture.** Inputs from different modalities $X = (X_1, ..., X_n)$ are first encoded by modality-specific encoders. Modality-specific features are processed by latent converters to map them into sequences of embeddings which are concatenated and fused by a transformer block. The output is a **single multimodal feature $\mathbf{Z}_\theta$**.

**Transformer block.** The goal of this module is to perfom the fusion of the modality-specific embeddings through the multihead self-attention layers in the transformer block, obtaining the final multimodal embedding $Z$.

More information about the specific modules used as modality encoders, about the latent converters and the transformer block architecture can be found in Appendix B.

## 3.3 TRAINING

Given a multimodal input $X = (X_1, ..., X_n)$ and a set of label-preserving multimodal transformations $\mathcal{T}^\star$, two augmentations $t', t''$ are drawn from $\mathcal{T}^\star$ and applied to $X$, obtaining $X' = t'(X)$ and $X'' = t''(X)$. We also consider projections (with a slight abuse of notation) $X_i = (\texttt{[MSK]}, ..., X_i, ..., \texttt{[MSK]})$ for $i \in \{1, ..., n\}$, where every modality is masked except for the $i$-th.

These terms are encoded by the network to obtain $(n+2)$ embeddings, namely: $Z', Z''$, and $\{Z_i\}_{i=1}^n$. $(2n+1)$ mutual information terms are then optimized through backpropagation: $I(Z', Z'')$ to maximize $I(Z, Z')$ in Eq. (3) and both $I(Z_i, Z')$ and $I(Z_i, Z'')$ to better approximate $R + U_i$ in Eq. (5) for $i \in \{1, ..., n\}$.

We use the InfoNCE (Oord et al., 2018) estimator of mutual information due to its simplicity and strong results in the self-supervised learning literature:

$$\hat{I}_{\text{NCE}}(Z, Z') = \mathbb{E}_{\substack{z, z'_{\text{pos}} \sim p(Z, Z') \\ z'_{\text{neg}} \sim p(Z')}} \left[ \log \frac{\exp \text{sim}(z, z'_{\text{pos}})}{\sum_{z'_{\text{neg}}} \exp \text{sim}(z, z'_{\text{neg}})} \right] \tag{6}$$

Given this estimator, our final training loss can be written as:

$$\mathcal{L}_{\text{CoMM}} = - \underbrace{\hat{I}_{\text{NCE}}(Z', Z'')}_{\approx R + S + \sum_{i=1}^n U_i} - \sum_{i=1}^n \underbrace{\tfrac{1}{2}\left( \hat{I}_{\text{NCE}}(Z_i, Z') + \hat{I}_{\text{NCE}}(Z_i, Z'') \right)}_{\approx R + U_i} =: \mathcal{L} + \sum_{i=1}^n \mathcal{L}_i \tag{7}$$

Fig. 3 illustrates CoMM's training process for the case $n = 2$. The pseudocode for the general case $n \geq 2$ is available in Appendix D. It is worth to notice that the loss terms in Eq. (7) grow *linearly* with the number of modalities $n$.

**At inference**, no augmentations are applied. The multimodal input $X = (X_1, ..., X_n)$ is processed through the network to obtain the multimodal feature $Z_\theta = f_\theta(X)$. This multimodal representation can then be directly transferred to any task either by performing linear probing or by fine-tuning the whole architecture.

## 4 EXPERIMENTS

We design different sets of experiments to evaluate our model: first, over a controlled environment inspired by the Trifeature dataset (Hermann & Lampinen, 2020), we carefully formulate tasks that

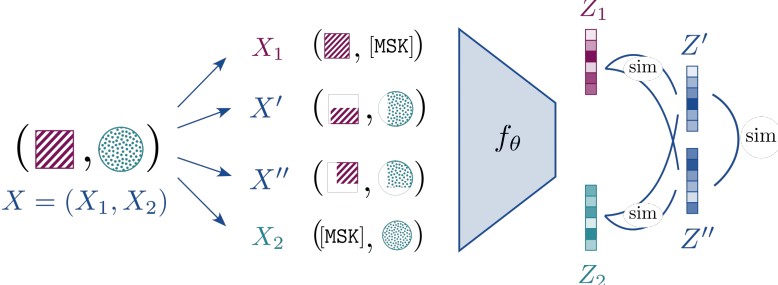

Figure 3: **CoMM training for** $n = 2$. Two multimodal augmentations are applied to $X$ to obtain $X'$ and $X''$. We also consider the projection operators to get $\{X_i\}_{i=1}^{n}$. These $n + 2$ transformed versions of $X$ are processed by the network $f_\theta$, trained to maximize the agreement between these $n + 2$ terms using contrastive objectives.

need a specific kind of interaction (uniqueness, redundancy, or synergy) to assess the model's ability to capture them separately. Second, in order to evaluate the capacities of the model to learn multimodal representations for complex tasks requiring different levels of shared, unique, and synergistic information, we use real-world benchmark datasets with 2 or 3 modalities.

**Experimental settings.** We report mean and standard deviation over 5 runs for all our results (except when using public model weights). All hyper-parameter details can be found in Appendix B.

**Evaluation.** Given a pre-trained model $f$, we perform evaluation through linear probing, *i.e.*, we train a linear layer $g_W(x) = W f(x)$ ($f$ fixed) to minimize the classification or regression loss (depending on the task). We also report results obtained with fine-tuning, *i.e.* after further training $f$ in a supervised way.

## 4.1 CONTROLLED EXPERIMENTS ON THE BIMODAL TRIFEATURE DATASET

To evaluate whether CoMM learns redundant, synergistic and unique information for a given task, we design a synthetic bimodal dataset based on Trifeature (Hermann & Lampinen, 2020). Briefly, we generate a Trifeature dataset (as first modality) containing images of one of ten shapes, for one of ten textures and one of ten colors (1 000 combinations in total). We augment each image three times using rotations and translations. Then, we pair each image with a second one (as second modality) from the same Trifeature dataset, allowing us to control the shared, unique and synergistic attributes between the two modalities. As Hermann & Lampinen (2020), we use AlexNet (Krizhevsky et al., 2012) as modality-specific encoder for both modalities in all experiments with an embedding dimension $d = 512$.

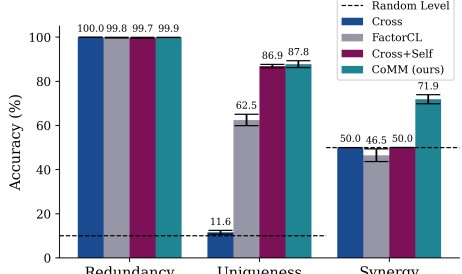

Figure 4: Linear probing accuracy of redundancy (shape), uniqueness (texture) and synergy (color and texture) on bimodal Trifeature dataset. CoMM is the only model capturing all three task-related interactions between modalities.

**Experiment 1 – Modeling shared and unique interactions.** We choose the shape as shared feature and texture as unique feature for each modality by selecting only paired images with the same shape. Training and test set follow the same distribution, with 10 000 and 4 096 images, respectively. We measure the linear probing accuracy of shape (respectively texture) to test whether redundancy (resp. uniqueness) has been captured in the latent representation of the model (chance level=10%).

**Experiment 2 – Modeling synergy.** In the previous experiment, texture and color features are independent between the two modalities. To introduce a synergy between these features, we bias the training set by defining a mapping $\mathcal{M}$ between the ten textures and ten colors (*e.g.* stripes=red, dots=green, *etc.*). Then, we select randomly 10 000 pairs of images that respect this mapping, thus artificially introducing a strong correlation between texture from the first modality and color from the second modality in the training set. The test set is left unchanged from previous Exper-

| Model | Regression | Classification | | | | |
|---|---|---|---|---|---|---|
| | V&T EE ↓ | MIMIC ↑ | MOSI ↑ | UR-FUNNY ↑ | MUsTARD ↑ | Average* ↑ |
| Cross† (Radford et al., 2021) | $33.09_{\pm 3.67}$ | $66.7_{\pm 0.0}$ | $47.8_{\pm 1.8}$ | $50.1_{\pm 1.9}$ | $53.5_{\pm 2.9}$ | 54.52 |
| Cross+Self† (Yuan et al., 2021) | $7.56_{\pm 0.31}$ | $65.49_{\pm 0.0}$ | $49.0_{\pm 1.1}$ | $59.9_{\pm 0.9}$ | $53.9_{\pm 4.0}$ | 57.07 |
| FactorCL† (Liang et al., 2023b) | $10.82_{\pm 0.56}$ | $\mathbf{67.3}_{\pm 0.0}$ | $51.2_{\pm 1.6}$ | $60.5_{\pm 0.8}$ | $55.80_{\pm 0.9}$ | 58.7 |
| CoMM (ours) | $\mathbf{4.55}_{\pm 0.30}$ | $66.4_{\pm 0.32}$ | $\mathbf{67.5}_{\pm 1.30}$ | $\mathbf{63.1}_{\pm 0.65}$ | $\mathbf{63.9}_{\pm 3.01}$ | $\mathbf{65.22}$ |
| SupCon† (Khosla et al., 2020) | - | $67.4_{\pm 0.0}$ | $47.2_{\pm 1.2}$ | $50.1_{\pm 2.0}$ | $52.7_{\pm 2.2}$ | 54.35 |
| FactorCL-SUP† (Liang et al., 2023b) | $1.72_{\pm 0.03}$ | $\mathbf{76.8}_{\pm 0.0}$ | $69.1_{\pm 0.6}$ | $63.5_{\pm 0.8}$ | $69.9_{\pm 1.9}$ | 69.82 |
| CoMM (fine-tuned) | $\mathbf{1.34}_{\pm 0.01}$ | $68.18_{\pm 0.23}$ | $\mathbf{74.98}_{\pm 0.43}$ | $\mathbf{65.96}_{\pm 0.44}$ | $\mathbf{70.42}_{\pm 0.15}$ | $\mathbf{69.88}$ |

Table 1: Linear evaluation top-1 accuracy (in %) for classification tasks and MSE ($\times 10^{-4}$) for regression task (V&T End Effector) on MultiBench after 100 epochs. †Results obtained from (Liang et al., 2023b). * Average is taken over classification results only. Rows in color are supervised.

iment 1 and the task $Y$ is to detect whether a given pair of images respects the mapping $\mathcal{M}$, *i.e.* $Y = \mathbb{1}(\text{texture}(X_1), \text{color}(X_2) \in \mathcal{M})$ (chance level=50%).

In Fig. 4, we show that cross-modality constraints with InfoNCE (Radford et al., 2021) ("Cross" model) allow to perfectly capture redundant information but completely remove unique and synergistic information, as predicted in Lemma 1. Self-supervised constraints on each encoder ("Cross+Self" (Yuan et al., 2021)) capture accurately unique information but fail at preserving synergy, FactorCL, the method most closely related to our work, also performs poorly on synergy. CoMM is the only model learning all three interactions related to the task.

## 4.2 EXPERIMENTS WITH 2 MODALITIES ON REAL-WORLD DATASETS

**MultiBench.** Following (Liang et al., 2023b), we use a subset of real-world multimodal datasets from MultiBench (Liang et al., 2021), with different degrees of shared and unique task-relevant information, including: *Vision&Touch* (Lee et al., 2020) a robotics dataset that includes images, force, and proprioception data for end-effector position prediction (regression) and contact prediction (binary classification), *MIMIC* (Johnson et al., 2016), a dataset for mortality and disease prediction from medical records, including tabular data and medical time series from ICU; *MOSI* (Zadeh et al., 2016), a dataset for sentiment analysis from videos (vision, audio, and language); *UR-FUNNY* (Hasan et al., 2019), humor detection from videos (vision, audio and language); and *MUsTARD* (Castro et al., 2019), a dataset for sarcasm detection from TV shows (vision, audio, and language). For fair comparisons, we run our experiments using the same data pre-processing steps as previous works, using pre-extracted text, video and audio features for training (Liang et al., 2021; 2023b). More details about these datasets and the data pre-processing can be found in Appendix E. We train CoMM on the same data modalities as FactorCL and use the same backbone networks.

We consider two different experimental settings. In the *self-supervised setting*, we perform pre-training and evaluate with linear probing. We consider FactorCL, "Cross" and "Cross+self" methods for comparison. In the *fine-tuning setting*, CoMM is fully fine-tuned in a supervised way after pre-training. We compare it against SupCon and FactorCL-Sup as supervised methods.

Results for these experiments are in Table 1. In the self-supervised experiments with linear probing evaluation, CoMM surpasses FactorCL (second best) by large margins (16.3%, 2.6% and 8.1% of top-1 accuracy on MOSI, UR-FUNNY and MUsTARD, respectively) on three out of four classification datasets. On MIMIC, margins are considerably narrower, with FactorCL performing slightly better, and CoMM showing comparable results with "Cross" and "Cross+Self". In the regression task of V&T, CoMM is considerably better than competitors ($3 \times 10^{-4}$ lower MSE than second best). In the fine-tuning scenario, we observe the same pattern, with CoMM outperforming competitors on four datasets and FactorCL taking the lead on MIMIC.

These experiments not only show CoMM's efficiency at learning multimodal representations, but also exhibit CoMM's versatility to process different data domains (time series, audio, images, text, *etc.*) and to adapt to diverse backbones.

**Multimodal IMDb** (MM-IMDb) (Arevalo et al., 2017) is a dataset for movie genre prediction. We consider two modalities: images and text (the movie poster and its plot's description, respectively). Since each movie can be classified into one or more genres, it is a multi-label classification task, with 23 categories. MM-IMDb provides a suitable example of life-like multimodal task as the genre prediction cannot be performed accurately from the movie poster or the movie plot alone, while results significantly improve by considering both (Arevalo et al., 2017), suggesting that synergistic interactions are needed to fulfill this task. We compare CoMM's performance on MM-IMDb against

| Model | Modalities | weighted-f1 | macro-f1 |
|---|---|---|---|
| SimCLR[†] (Chen et al., 2020a) | V | $40.35_{\pm 0.23}$ | $27.99_{\pm 0.33}$ |
| | V | 51.5 | 40.8 |
| CLIP (Radford et al., 2021) | L | 51.0 | 43.0 |
| | V+L | 58.9 | 50.9 |
| BLIP-2 (Li et al., 2023) | V+L | 57.4 | 49.9 |
| SLIP[†] (Mu et al., 2022) | V+L | $56.54_{\pm 0.19}$ | $47.35_{\pm 0.27}$ |
| CLIP[†] (Radford et al., 2021) | V+L | $54.49_{\pm 0.19}$ | $44.94_{\pm 0.30}$ |
| CoMM (ours, CLIP backbone) | V+L | $\underline{61.48}_{\pm 0.18}$ | $\underline{54.63}_{\pm 0.22}$ |
| CoMM (ours, BLIP-2 backbone) | V+L | $\mathbf{64.75}_{\pm 0.17}$ | $\mathbf{58.44}_{\pm 0.43}$ |
| MFAS (Pérez-Rúa et al., 2019) | V+L | 62.50 | 55.6 |
| ReFNet (Sankaran et al., 2022) | V+L | - | 56.7 |
| CoMM[‡] (ours, CLIP backbone) | V+L | $\underline{64.90}_{\pm 0.21}$ | $\underline{58.97}_{\pm 0.19}$ |
| CoMM[‡] (ours, BLIP-2 backbone) | V+L | $\mathbf{67.39}_{\pm 0.07}$ | $\mathbf{62.0}_{\pm 0.25}$ |
| LLaVA-NeXT (Li et al., 2024) | V+L | 64.28 | 56.51 |

Table 2: Linear evaluation F1-score (weighted and macro) (in %) on MM-IMDb after 70 epochs. † indicates further training on unlabeled data. ‡ means supervised fine-tuning. Rows in color are supervised.

important baselines under two different settings. First, in the *self-supervised setting*, we consider CLIP (Radford et al., 2021), representing *"Cross"* methods, SLIP (Mu et al., 2022) representing *"Cross+Self"* methods, SimCLR (Chen et al., 2020a) for unimodal self-supervised methods, BLIP-2 (Li et al., 2023) as a recent powerful vision and language model, as baselines. All models were trained on unlabeled MM-IMDb. For CLIP, we also report results with the publicly released weights without further training. CoMM is initialized with pre-trained weights from CLIP and BLIP-2. Second, in the *fine-tuning setting*, we compare CoMM fine-tuned with state-of-the-art fully supervised baselines. We also include results for LLaVA-NeXT (Li et al., 2024) representing new vision-and-language generative models, however these scores are not fully comparable to ours, since the model might have seen the IMDb database during training. See Appendix B for implementation details.

Table 2 shows that CoMM outperforms all models in both settings. In the self-supervised setting, CoMM has a margin of 7.5% and 5.8% of macro and weighted F1-scores, respectively, with the second-best method. It is interesting to observe that CLIP performs better with the publicly released weights (probably because of the large-scale and diversity of the data it was trained on), than with further training on MM-IMDb. This result suggests that the genre movie prediction does not benefit from learning redundant information. Including uniqueness from the image modality allows for some improvement (SLIP). In the fine-tuning setting, CoMM fine-tuned outperforms existing fully supervised baselines, even if MFAS has been designed to search for the best fusion strategy, and vision-language generative models (LLaVA-NeXT).

## 4.3 EXPERIMENTS WITH 3 MODALITIES ON REAL-WORLD DATASETS

We test CoMM's abilities to learn multimodal interactions beyond 2 modalities. We perform experiments on two large datasets including tri-modal data from MultiBench: *Vision&Touch* (contact prediction task) and *UR-FUNNY*.

In Table 3, we compare CoMM trained on the three modalities in a self-supervised way against CMC (Tian et al., 2020a). We also compare with bi-modal models: CoMM, *Cross* and *Cross+Self* trained on image and proprioception data for Vision&Touch and image and text data for UR-FUNNY.

| Model | #Mod. | V&T CP | UR-FUNNY |
|---|---|---|---|
| Cross | 2 | 84.4 | 50.1 |
| Cross+Self | 2 | 86.8 | 59.9 |
| CoMM (ours) | 2 | $\underline{88.1}$ | $\underline{63.1}$ |
| CMC (Tian et al., 2020a) | 3 | 94.1 | 59.2 |
| CoMM (ours) | 3 | $\mathbf{94.2}$ | $\mathbf{64.6}$ |

Table 3: Linear evaluation top-1 accuracy (%) on Vision&Touch and UR-FUNNY.

First, we observe a consistent improvement (+6.1% on V&T, +1.5% on UR-FUNNY) when adding a third modality with CoMM (compared to only using two), which demonstrates its versatility. Second, we improve the state-of-the-art for SSL methods on datasets with more than two modalities (+0.1% and +5.4% on V&T and UR-FUNNY, respectively).

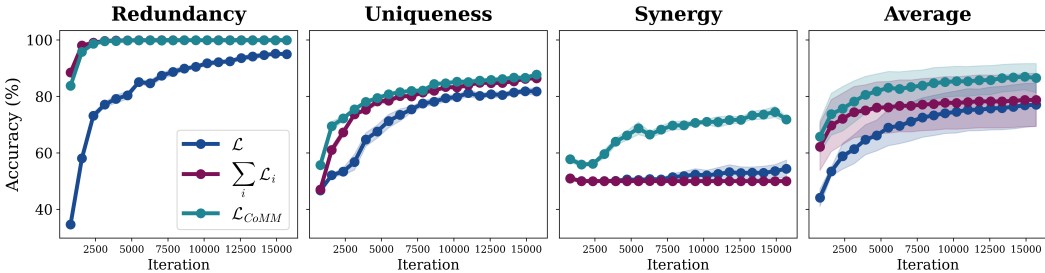

Figure 5: Linear probing accuracy of redundancy $R$, uniqueness $U = \frac{1}{n}\sum_{i=1}^{n} U_i$ and synergy $S$ on bimodal Trifeature when optimizing each term separately in $\mathcal{L}_{\text{CoMM}}$. Minimizing $\mathcal{L}_i$ allows to learn $U_i$ and $R$, approximating $I(X_i; Y)$ for $i \in \{1, ..., n\}$. Optimizing $\mathcal{L} = -\hat{I}(Z', Z'')$ allows to slowly learn $R$, $U_i$ and $S$. CoMM quickly captures all information.

## 5 ABLATION STUDIES

We test three main components of our framework (the loss, fusion module and augmentation strategy) against important control baselines on bimodal Trifeature dataset (see Section 4.1).

**Loss function.** First, we check our claim that optimizing both $\mathcal{L}$ and $\sum_{i=1}^{n} \mathcal{L}_i$ –as in Eq. (7)– is required to accurately capture uniqueness, synergy, and redundancy. In Fig. 5, we show that minimizing $\sum_{i=1}^{n} \mathcal{L}_i$ improves redundancy and uniqueness, as guaranteed by our Lemma 3, but fails for synergy. Conversely, minimizing $\mathcal{L}$ allows one to learn all information terms but very slowly. In particular, for synergy, we argue this is because the model has to learn modality-specific features first (phase 1) before learning their interactions (phase 2). The former is learned through $I(Z_i, Z') + I(Z_i, Z'')$ while the latter is captured with $I(Z', Z'')$. Hence, $\mathcal{L}_{\text{CoMM}}$ speeds up phase 1 and can learn synergy more efficiently in phase 2.

**Fusion module.** We compare our attention-based latent fusion module with shallow linear fusion. We project modality-specific representations to the common latent space using only linear layers and remove latent converters. Table 4 shows that synergy is not captured with linear fusion. This is expected as the XOR gate (typical example of synergistic interactions) cannot be approximated by a linear function. Additionally, uniqueness accuracy is also degraded compared to CoMM ($-6\%$), suggesting that only modeling linear interactions limits the model's representation power.

| Fusion | Redundancy | Uniqueness | Synergy | Average |
|---|---|---|---|---|
| Concat + Linear | $99.71_{\pm 0.06}$ | $81.49_{\pm 2.88}$ | $50.0_{\pm 0.0}$ | $77.07$ |
| CoMM | $\mathbf{99.92}_{\pm 0.03}$ | $\mathbf{87.83}_{\pm 1.55}$ | $71.87_{\pm 2.06}$ | $\mathbf{86.83}$ |

Table 4: Linear fusion module versus attention-based fusion with latent converters (CoMM). Non-linearities are required to learn synergistic interactions between modalities.

**Data augmentation.** Finally, we show in Table 5 that strong data augmentation is crucial for learning multi-modal interactions, in line with the literature on unimodal contrastive methods. Contrary to FactorCL (Liang et al., 2023b), applying strong augmentation on *both* modalities is beneficial for CoMM and we do not require task-dependent augmentations, highlighting the versatility of our framework.

| Augmentations | | $R$ | $U_1$ | $U_2$ | $S$ | Average |
|---|---|---|---|---|---|---|
| Modality 1 | Modality 2 | | | | | |
| {All} | ∅ | $99.72_{\pm 0.04}$ | $79.92_{\pm 1.05}$ | $46.44_{\pm 3.74}$ | $50.0_{\pm 0.0}$ | $69.02$ |
| ∅ | {All} | $99.58_{\pm 0.13}$ | $53.89_{\pm 6.69}$ | $86.04_{\pm 2.01}$ | $50.0_{\pm 0.0}$ | $72.37$ |
| {All}\{crop} | {All} | $88.79_{\pm 0.96}$ | $25.65_{\pm 0.10}$ | $84.00_{\pm 4.43}$ | $50.0_{\pm 0.0}$ | $62.11$ |
| {All} | {All}\{crop} | $90.50_{\pm 2.07}$ | $83.22_{\pm 2.86}$ | $21.74_{\pm 1.36}$ | $50.0_{\pm 0.0}$ | $61.36$ |
| CoMM | | $\mathbf{99.92}_{\pm 0.03}$ | $84.35_{\pm 2.37}$ | $\mathbf{91.19}_{\pm 0.97}$ | $\mathbf{71.87}_{\pm 2.06}$ | $\mathbf{86.83}$ |

Table 5: Effect of data augmentation on linear probing accuracy (%) of multimodal interactions. *All* refers to SimCLR augmentations (Chen et al., 2020a). CoMM uses *All* for both modalities.

## 6 RELATED WORK

**Multimodal learning** refers to methods that connect and integrate information from multiple sources of data (Baltrušaitis et al., 2018; Akkus et al., 2023). Early works focused on training

separate encoders and studied different fusion mechanisms to blend the information from different inputs (Zeng et al., 2007; Pérez-Rúa et al., 2019). With the development of transformers and ViTs, the focus has shifted towards training a unique architecture and designing specific tasks and loss functions to integrate multimodal interactions (Xu et al., 2023; Lu et al., 2019; Sun et al., 2019; Chen et al., 2020b; Lu et al., 2020). Today multimodal learning includes different research lines, from generative multimodal learning (Suzuki & Matsuo, 2022; Ramesh et al., 2021; Saharia et al., 2022; Wang et al., 2022b; Alayrac et al., 2022) to multimodal representation learning (Zong et al., 2024; Tian et al., 2020a). CoMM belongs to the latter category.

**Self-supervised multimodal representation learning.** Self-supervised learning aims to learn general representations through supervisory signals from the data itself (known as the pretext task) (Balestriero et al., 2023). In the multimodal context (Zong et al., 2024), self-supervised methods can be grouped according to their pretext task. Clustering-based methods (Alwassel et al., 2020; Asano et al., 2020), where cluster assignments are used as pseudo-labels to train the model and different modalities can be used as supervisory signals to each other; masking modeling methods (Mizrahi et al., 2023; Bachmann et al., 2022; Lu et al., 2023) that reconstruct pieces of information that have been masked from input data. In the multimodal case, masked modelling is used in a cross-modal way, by predicting missing information conditioned in other modalities. Contrastive methods (Radford et al., 2021; Jia et al., 2021) in the multimodal context have been mostly used on matched data from different modalities to obtain aligned–yet distinct–representations. In this work, instead of representing each modality separately, we take a different perspective for contrastive methods by learning a single multimodal representation of the inputs.

**Multimodal contrastive learning.** Current methods in contrastive learning are inspired by the idea of multiview learning (Li et al., 2018) and have shown remarkable results in representation learning in unimodal tasks (Chen et al., 2020a; He et al., 2020; Tian et al., 2020a; Oord et al., 2018). The natural extension of these approaches to multimodal settings is to optimize a cross-modal contrastive loss (Radford et al., 2021; Alayrac et al., 2020; Jia et al., 2021). Other works have gone even further by introducing cross-modal and intra-modal contrastive objectives (Mu et al., 2022; Jain et al., 2021; Mi et al., 2024), or by adding other intra-modality regularization terms (Wang et al., 2022a; Kim et al., 2022). However, these approaches are designed to learn redundant information, neglecting the contributions of uniqueness or synergy. Recently, FactorCL (Liang et al., 2023b) has proposed a solution to model shared and unique task-relevant information explicitly. Yet, the method relies heavily on assumptions that are hard to meet in practice; it proposes a factorized approximation of multimodal interactions that is prone to cumulative errors and does not model synergistic information. Alternatively, we propose CoMM, a contrastive multimodal approach that leverages multimodal augmentations with a modular architecture optimized through information theory-grounded losses to capture unique, redundant and synergistic interactions.

## 7 CONCLUSIONS

Multisensory integration is at the core of human perception, allowing us to build coherent representations of our environment. In this paper, we introduce CoMM, a contrastive multimodal method that enables the integration of multiple modalities in a single multimodal representation space. Unlike existing multimodal contrastive models, CoMM is designed to learn multimodal interactions beyond redundancy, through Partial Information Decomposition theory. Our controlled experiments on the bimodal Trifeature dataset demonstrate that CoMM successfully learns redundant, unique and synergistic information. In real-life multimodal datasets from Multibench with two and three input modalities, CoMM outperforms existing methods by large margins in almost every case, showing the efficiency and versatility of CoMM to handle data across diverse domains (robotics, healthcare, affective computing, multimedia) and data structures (time series, image, text, audio, tabular).

This work opens large avenues for future research on multimodal representation learning, in particular for crafting label-preserving multimodal augmentations not limited to unimodal augmentations. Limitations and perspectives for future research are further discussed in Appendix A. Overall, the simplicity and versatility of CoMM's design make it a good candidate to learn deep representations of several modalities across domains. It offers promises in better solving real-world problems ranging from neuroscience (Preti & Van De Ville, 2019) and medical imaging (Boehm et al., 2022) to remote sensing (Gómez-Chova et al., 2015).

ACKNOWLEDGMENTS

We thank Jonathan Sauder, Valentin Gabeff and Valérie Zermatten for providing helpful feedback on earlier versions of this work. JCN acknowledges the support from EPFL Science Seed Fund. BD acknowledges the support from the PHRT project number 643.

ETHICS STATEMENT

We acknowledge that there exist potential privacy risks when using human behavior or medical data. However, we minimize these potential risks by using publicly available datasets that have been carefully collected, respecting participants' consent, de-identifying medical data and anonymizing video data. In this work, we have followed best practices in maintaining the privacy and safety of these datasets.

REPRODUCIBILITY STATEMENT

To ensure the reproducibility of our work, we have used publicly available datasets from Multi-Bench (Liang et al., 2021) (MIMIC, MOSI, UR-FUNNY, MUsTARD and Vision&Touch); MM-IMDb (Arevalo et al., 2017), and the synthetic Trifeatures dataset (Hermann & Lampinen, 2020). Details about these datasets and pre-processing steps can be found in Appendix E. Appendix B thoroughly describes neural network architectures, data augmentation strategies, and experimental settings used in all our experiments. We include in Appendix D a pseudo-code for implementing CoMM's training, and we released our code in this Github repository. We also provide details on processing times and model complexity compared to other multimodal models in Appendix F. Finally, all proofs for our theoretical developments can be found in Appendix G.

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

## A  LIMITATIONS AND FUTURE RESEARCH

- **CoMM's theoretical analysis for more than two modalities** is still unclear since PID theory is currently limited to $n = 2$ modalities (see Section 2 in the main paper). While uniqueness can be easily defined for any $n \geq 2$, the number of interactions between modalities (*e.g.* redundancy and synergy) grows exponentially with $n$, making the analysis harder. Nonetheless, we should recall that CoMM performs empirically very well even for $n > 2$ (see Section 4.3), and the number of loss terms increases only linearly with $n$ (see Eq. (7)). Another limitation is the additional computational cost associated with adding modality-specific encoders. A simple workaround is to use large pretrained (frozen) encoders (*e.g.* from CLIP (Radford et al., 2021) for vision and text or DINOv2 (Oquab et al., 2024) for vision) and to only tune a lightweight fusion transformer in CoMM, allowing a much faster training.

- **CoMM's computational cost for data augmentation** is higher than in cross-modalities frameworks (such as CLIP). This is because for each data tuple $X = (X_1, X_2, \ldots, X_n)$, CoMM needs to compute $X' = t(X)$, while cross-modality methods feed $X$ directly into the neural network. A possible solution would be the implementation of a momentum encoder and a queue method, as in MoCo (He et al., 2020).

- **Interpretability.** Our experiments on the Trifeature dataset (see Section 4.1 in the main paper) show that CoMM can efficiently learn unique, redundant and synergistic information. However, it seems difficult to disentangle the contributions of these three interactions in the representation space. Disentanglement might be one direction for future work. However, other approaches to measure such quantities (given a dataset and a task) are emerging (Hu et al., 2022; Liang et al., 2023a; 2024). Another interesting approach would be to use modality masking (already implemented and handled by CoMM) to analyze the contribution of each modality individually versus collectively.

We believe the above limitations are directions that were outside the scope of the current manuscript; however, they are exciting avenues for future research in multimodal representation learning.

## B  IMPLEMENTATION DETAILS

For all experiments with CoMM, we use an attention-based fusion module that takes as input a sequence of embeddings. We use a 1-layer Transformer with 8 heads that applies self-attention to all inputs and we add a `[CLS]` learnable embedding at the beginning of the sequence. The embedding size depends on each dataset, detailed below.

### B.1  ENCODER ARCHITECTURES BY DATASET

- **Trifeature** is composed of two visual modalities, so we use an AlexNet encoder for both modalities with a 512-d embedding space. For CoMM, we remove the last average pooling layer and we apply a linear patch embedding layer (Dosovitskiy et al., 2021) to the $6 \times 6$ feature maps as latent converter. We then add fixed 2D sine-cosine positional embeddings (Dosovitskiy et al., 2021). For *Cross*, we use a linear projector for each encoder (512-d output space) and we optimize the CLIP loss (Radford et al., 2021). For *Cross+Self*, we apply a 3-layers MLP projection head to each encoder to optimize the SSL objectives as in SLIP (Mu et al., 2022) (1024-d hidden layers and 256-d output space) and we use the same projectors and CLIP objective as for *Cross*.

- **MIMIC** contains tabular and time-series data, seen as two modalities. We use a 2-layers MLP (10-d hidden dimension, 10-d output) as tabular encoder and a GRU (512-d hidden dimension) as time-series encoder (similarly to FactorCL (Liang et al., 2023b)). For tabular data, we use a feature tokenizer (Gorishniy et al., 2021) as latent converter with 512-d embedding space and no converter for time-series.

- **MOSI**, **UR-FUNNY** and **MUsTARD** contain visual and textual modalities extracted from videos. Similarly to FactorCL (Liang et al., 2023b), we use a 5-head Transformer with 5 layers for each modality with a 40-d embedding space. We do not use latent converters in this case.

- **MM-IMDb** also contains visual and textual modalities but in their raw format. We use a ViT-B/32 (Dosovitskiy et al., 2021) image encoder pre-trained with CLIP (Radford et al., 2021) and a Sentence-BERT multilingual text encoder[2] pre-trained with CLIP and distilled with Sentence-BERT (Reimers & Gurevych, 2019). For CoMM, we consider the token embeddings given by the image and text encoders (frozen) and we do not use latent converters. For CLIP (Radford et al., 2021), we fine-tune the pre-trained encoders with their original architecture. For SLIP (Mu et al., 2022), we use the same pre-trained encoders as CLIP and we add a 3-layers visual projection MLP (4096-d hidden layers and 256-d output space) to compute the SSL objectives.

- **Vision&Touch** has visual, force-torque and robot proprioception modalities available. For the binary contact prediction task, we only use visual and proprioception data for the experiments with two modalities and we encode images with a ResNet18 (He et al., 2016) (512-d output space) and proprioception data with a 5-layer MLP (512-d output space), as in the original paper (Lee et al., 2020). In the experiments with 3 modalities, force-torque data are encoded with a 5-layer causal convolutions network (512-d output space). For CoMM, we remove the last average pooling layer and apply a patch embedding layer (Dosovitskiy et al., 2021) to the $4 \times 4$ feature maps as latent converter. We consider the 1D feature vector of proprioception data as a 1-length sequence in the fusion module. For the end-effector regression task, we use visual and force-torque modalities. We use ResNet18 as image encoder (128-d output space) and a 5-layer causal convolutions network (128-d output space) as force encoder (Lee et al., 2020). For CoMM, we add the same latent converter for images as in the previous task and we add a feature tokenizer (Gorishniy et al., 2021) for force-torque embeddings.

## B.2 Data augmentation by modality

For raw images (in Trifeature, MM-IMDb and Vision&Touch), we use the default SimCLR augmentations (Chen et al., 2020a), which include *RandomResizedCrop*, *ColorJitter*, *RandomGrayscale*, *GaussianBlur* and *RandomHorizontalFlip* (from the PyTorch library).

For tabular data (in MIMIC and Vision&Touch), we add a random Gaussian noise to each component (assuming they are all continuous).

For time-series data (either extracted from videos as in MOSI, UR-FUNNY, MusTARD, from health recordings as in MIMIC or from force-torque readings as in Vision&Touch), we apply a random composition of Gaussian noise and random dropping between 0 and 80% of the sequence. We have compared several other strategies for this modality and we present the results in Appendix C.

For raw text (in MM-IMDb), we randomly mask 15% of input tokens by using a special `[MASK]` token as in BERT (Devlin et al., 2019).

## B.3 Latent converters by encoder architecture

- Transformer and GRU: the latent converter is the identity since the Transformer and GRU already output a sequence of embeddings;
- CNN: the latent converter is a patch embedding projection module originally defined in ViT (Dosovitskiy et al., 2021) that we apply to the features maps of the CNN;
- MLP: the latent converter is a feature tokenizer originally defined in (Gorishniy et al., 2021) for tabular data. The feature vector $h_i$ is transformed into sequential embeddings by applying feature-wise multiplication with a learnable matrix and we add a bias term. For proprioception and force-torque data, we simply consider them as a 1-length sequence in the fusion module.

## B.4 Experimental settings

We use AdamW optimizer (Loshchilov & Hutter, 2019) in all experiments and a learning rate $\alpha = 3 \times 10^{-4}$ for Trifeature (with weight decay $10^{-4}$), $\alpha = 10^{-3}$ for MIMIC, MOSI, UR-FUNNY and MusTARD (with weight decay $10^{-2}$) and $\alpha = 10^{-4}$ for MM-IMDb and Vision&Touch (with weight decay $10^{-2}$). For MM-IMDb, we also use a cosine scheduler with final value $10^{-6}$ and a warmup over 10 epochs. All models were optimized during 100 epochs. The *critic* in the InfoNCE

---

[2]https://huggingface.co/sentence-transformers/clip-ViT-B-32-multilingual-v1

losses in $\mathcal{L}_{\text{CoMM}}$ Eq. (7) is implemented as a 3-layer MLP (512-d hidden layers and 256-d output space), similarly to the projection head in SimCLR (Chen et al., 2020a). All experiments ran on a single V100 GPU with 32GB of memory.

**Fine-tuning of CoMM.**   For all downstream classification tasks, we use the SupCon loss (Khosla et al., 2020) to fine-tune CoMM (with no additional parameters). In the case of multi-label classification with MM-IMDb, we use a linear head on top of CoMM and we optimize a cross-entropy loss for each label. For the regression task on Vision&Touch, we also use a linear head and we optimize the MSE loss. We systematically use early-stopping according to the validation accuracy in order to prevent over-fitting on the downstream tasks.

### B.5   EXPERIMENTAL SETTINGS ON THE BIMODAL TRIFEATURE DATASET

To generate our trifeature dataset, we considered the $1\,000$ combinations of the three features existing in the original dataset (see Appendix E) and split them into 800 combinations for training and 200 for evaluation. To have more variety in the training set for training, each combination was generated 3 times (the shape and the texture were randomly rotated), obtaining a training split of $2\,400$ images.

The bimodal Trifeature dataset used in our experiments was built by considering the trifeature dataset twice (as two separate modalities) and building pairs from these two dataset copies. In total, we get $5\,760\,000$ pairs ($2\,400 \times 2\,400$) available for training, and $40\,000$ ($200 \times 200$) available for evaluation.

To create a controlled environment for evaluation of multimodal interaction learning, we needed to carefully design tasks where the dominant interaction was clearly defined.

1. To measure uniqueness $U_1$ (resp. $U_2$), given a pair of trifeature images, the task is to predict the texture of the first (resp. the second) image. The task is then a 10-class classification problem and chance level is at 10%.

2. To measure redundancy $R$, given a pair of trifeature images with the same shape (but different color and texture), the task is to predict the shape of the pair (therefore, the redundant information that can be extracted either from the first or the second image). The task is then a 10-class classification problem and chance level is at 10%.

3. To measure synergy $S$, the definition of the task was more subtle as it should require information from both modalities simultaneously and should not be possible to perform it from one of the images alone. To achieve this, we defined a mapping $\mathcal{M}$ between the ten textures and the ten colors (e.g. stripes=red, dots=green, etc.). Then, given a pair of trifeature images, the task is to predict whether the pair satisfies the mapping or not. The task is then a binary classification problem and chance level is at 50%.

To evaluate these tasks, we built two versions of the bimodal trifeature dataset:

(i) For uniqueness and redundancy, we considered $10\,000$ image pairs (out of the $5\,760\,000$ pairs) for training and $4\,096$ for testing, that have the same shape (to measure redundancy) and different texture (to measure uniqueness).

(ii) For synergy, we considered $10\,000$ image pairs that respect the mapping $\mathcal{M}$ and used the same test set as before ($4\,096$ image pairs).

### B.6   ADDITIONAL DETAILS FOR LLAVA-NEXT EVALUATION

We evaluate LLaVA-NeXT (Li et al., 2024) on MM-IMDb based on its answer to the following prompt: "*From the following plot:* {...} *and this poster image, give me all the movie genres it belongs to among the following list:* {...}. *Give me the answer as a list.*" We formulate it as a close question with limited number of answers to be closer to the linear probing setting for representation learning models.

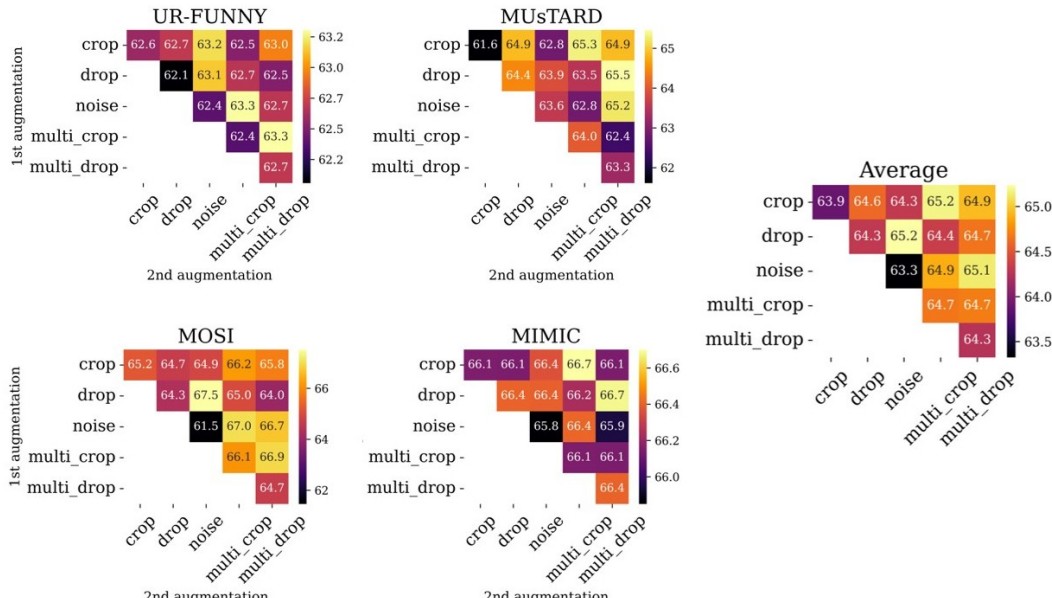

Figure 6: Linear evaluation accuracy (%) on four datasets from MultiBench (Liang et al., 2021) when applying individual or composing data augmentations on the modalities. Diagonal entries correspond to one augmentation applied to all modalities while off-diagonal entries are a composition of two augmentations. *Average* is the average of all four matrices. Results are averaged over 5 independent runs (with different seeds) in each cell.

## C ADDITIONAL EXPERIMENTS

In this section we present additional analysis of CoMM, including: a benchmark of several augmentation strategies for time-series data, ablation studies on our loss function on a real dataset, and supplementary results on the trifeatures dataset.

### C.1 DATA AUGMENTATION STRATEGY FOR TIME-SERIES

For time-series data like in MIMIC, MOSI, UR-FUNNY and MuSTARD, there is no consensus with respect to the best data augmentation strategy to apply for self-supervised contrastive learning. We benchmark several common augmentations for time-series data including random crop between 8% and 100% of the signal, random drop between 0% and 80% of the signal and adding Gaussian noise. We also designed two new multimodal augmentations: *multi-crop* (resp. *multi-drop*) consisting of cropping (resp. dropping) a signal across multiple modalities for time-aligned data (introducing consistency in the preserved multimodal signal).

We tested these augmentations along with their composition on four datasets from Multi-Bench (Liang et al., 2021) and we plot the results in Fig. 6. Overall, we find that composing Gaussian noise and random drop results in the best performances across datasets and tasks. This is our default augmentations strategy in our main experiments. Our proposed multi-drop and multi-crop augmentations can provide better results in some cases (for MIMIC and MuSTARD), but we select the same default augmentations for consistency across all datasets.

### C.2 ABLATION STUDIES ON A REAL DATASET

We performed the loss ablation study on MM-IMDb. Results follow the same tendency as in the Tri-features dataset (Fig. 5 main paper).

| Loss | weighted-f1 | macro-f1 |
|---|---|---|
| $\sum_i \mathcal{L}_i$ | $60.71_{\pm 0.17}$ | $53.35_{\pm 0.37}$ |
| $\mathcal{L}$ | $54.94_{\pm 0.50}$ | $47.13_{\pm 0.56}$ |
| $\mathcal{L}_{\text{CoMM}} = \mathcal{L} + \sum_i \mathcal{L}_i$ | $\mathbf{61.48}_{\pm 0.18}$ | $\mathbf{54.63}_{\pm 0.22}$ |

Table 6: Ablation study of loss function contributions on MM-IMDb. $\mathcal{L}_{\text{CoMM}}$ allows to better capture multimodal interactions than each term separately.

## C.3 DESIGN CHOICES ON TRIFEATURES EXPERIMENTS

Hermann & Lampinen (2020) experimented with Trifeatures using AlexNet and ResNet-50 backbones. Both architectures showed comparable results. Therefore, we chose to use AlexNet in our experiments. For completeness, we ran the same experiments on ResNet-50, which show that CoMM is the only model to learn all interactions, regardless of the architecture.

| Model | Redundancy | | | Uniqueness | | | Synergy | | |
|---|---|---|---|---|---|---|---|---|---|
| | Cross | C+S | CoMM | Cross | C+S | CoMM | Cross | C+S | CoMM |
| AlexNet | $100.0_{\pm 0.02}$ | $99.7_{\pm 0.2}$ | $99.9_{\pm 0.03}$ | $11.6_{\pm 0.9}$ | $86.9_{\pm 0.8}$ | $87.8_{\pm 1.6}$ | $50.0_{\pm 0.0}$ | $50.0_{\pm 0.03}$ | $71.9_{\pm 2.0}$ |
| ResNet | $100.0_{\pm 0.04}$ | $99.9_{\pm 0.04}$ | $99.9_{\pm 0.03}$ | $6.5_{\pm 0.7}$ | $96.2_{\pm 0.8}$ | $96.3_{\pm 1.3}$ | $50.0_{\pm 0.0}$ | $50.0_{\pm 0.0}$ | $75.0_{\pm 1.7}$ |

Table 7: Linear probing accuracy of redundancy (shape), uniqueness (texture) and synergy (color and texture) on bimodal Trifeature dataset, with different backbone encoders. These results are complementary to Fig. 4.

## C.4 ON THE FEASIBILITY OF ASSUMPTION 1

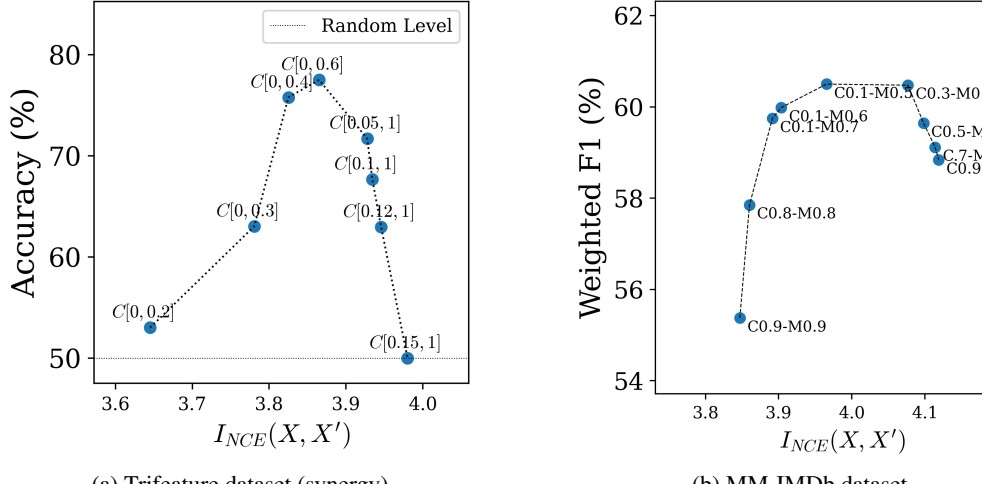

(a) Trifeature dataset (synergy)    (b) MM-IMDb dataset

Figure 7: Downstream task performance as the cropping augmentation strength decreases (for Trifeatures) or when both cropping and masking augmentation strength for image and text respectively decreases. $I_{NCE}(X, X')$ is measured using the trained encoder $f_\theta$ in Eq. (6). As $I_{NCE}(X, X')$ increases, the downstream performance first increases and then decreases, revealing a "sweet spot". It suggests the existence of an optimal augmentation policy that preserves task-relevant information while removing nuisance features from the multimodal input data, giving more credit to our Assumption 1.

Assumption 1 states the existence of an optimal augmentation policy $\mathcal{T}^*$ such that $I(X, t(X)) = I(X, Y)$ for any $t \in \mathcal{T}^*$, given a task $Y$. Hence, the set of optimal multimodal transformations

is task-dependent. However, in practice, we have seen that the augmentations applied in our experiments are general enough to obtain good results in a variety of datasets (7 datasets, 10 tasks, diverse data modalities). These augmentations were chosen according to the findings in unimodal self-supervised learning [2, 3], and by a comprehensive ablation study on the best augmentation strategies applicable to time-series in Appendix C.1.

Theoretically, the set of multimodal augmentations need to be large enough such that no information about the task $Y$ is lost, but small enough to extract this information only. Tian et al. (2020b) have referred to this observation as the **InfoMin principle**. In Section 5, our data augmentation ablation study shows that if the set of augmentations is not large enough, then synergy cannot be learnt (the performance is always random chance). In order to further explore the feasibility of Assumption 1 and inspired by the strategy developed by Tian et al. (2020b), we evaluate CoMM by progressively increasing the strength of the data augmentations applied. We use random crop as the augmentation to control in the vision domain, mainly for two reasons: first, it is intuitive that by decreasing the level of cropping (keeping less information about the images), we are destroying the task-relevant information; and second, because it has been empirically demonstrated that cropping is a critical transformation in self-supervised learning for vision Chen et al. (2020a). For the text domain, we use masking as the augmentation to control. More specifically, on Trifeatures we randomly crop the original image in the first modality from 0% up to $x\%$ ($x = 20\%$ is the strongest augmentation while $x = 60\%$ is the lightest); and from $x\%$ to 100% ($x = 0.05\%$ is the strongest, $x = 15\%$ the lightest). For MM-IMDb, we also use random crop of the image modality from $x\%$ up to 100% and masking of text with a decreasing probability $x\%$ from 90% (the strongest) to 20% (the lightest).

Our results are shown in Fig. 7. They demonstrate, both in the controlled environment of the bimodal Trifeatures dataset and in the real-world application of MM-IMDb, that the sweet spot of the InfoMin principle can be reached. By gradually increasing the strength of the applied transformations, we enhance model performance by reducing noisy information, up to an optimal point (the sweet spot) where noise is minimized while task-relevant information is preserved. However, applying overly strong augmentations destroys task-relevant information, leading to a degradation in model performance.

## D  PSEUDO-CODE

Algorithm 1 presents the pseudo-code for CoMM's training. It is written in the general case when we have $n$ modalities. It is complementary to Fig. 3 (main paper), which depicts the case for $n = 2$. CoMM's official implementation is available in this Github repository.

## E  DATASETS DETAILS

### E.1  TRIFEATURE

The Trifeature dataset (Hermann & Lampinen, 2020) was introduced as a controlled environment to study the properties of vision neural networks and how they learn different features (shape, texture and color). Images contain one of ten shapes (triangle, square, circle, *etc*.), rendered in one of ten textures (solid, stripes, grid, *etc*.), in one of ten colors (red, green, blue, *etc*.). Shapes are rendered within a $128 \times 128$ square, rotated at an angle drawn between $[-45°, 45°]$ and placed at a random position within a larger image ($224 \times 224$), such that the shape is fully contained in the image. Then, an independently rotated texture and a color are applied.

In our experiments, we consider the bimodal version of this dataset as explained in Section 4.1 (main paper).

### E.2  MULTIBENCH

All the following datasets are pre-processed as described in (Liang et al., 2021).

- **MIMIC** (*Medical Information Mart for Intensive Care*) (Johnson et al., 2016) is a dataset comprising de-indentified clinical data related to patients admitted to critical care units at a large

---

**Algorithm 1** CoMM training algorithm

---

**Require:** Multi-modal dataset $\{X_1, X_2, ..., X_n\}$, label-preserving transformations $\mathcal{T}^\star$, set of projection transformations $\mathcal{T} = \{t_1, \ldots, t_n\}$, batch size $N$, uni-modal encoders $(f_i)_{i \in [1..n]}$, fusion transformer $g$

**for** sampled mini-batch $\{\mathbf{x}_k\}_{k \in [1..N]} = (\mathbf{x}_k^1, ..., \mathbf{x}_k^n)_{k \in [1..N]}$ **do**

    **for** $k \in [1..N]$ **do**

        draw $t', t'' \sim \mathcal{T}^\star$

        $\mathbf{x}'_k, \mathbf{x}''_k \leftarrow t'(\mathbf{x}_k), t''(\mathbf{x}_k)$

        $\mathbf{z}'_k \leftarrow g(f_1(\mathbf{x}'^1_k), ..., f_n(\mathbf{x}'^n_k))$

        $\mathbf{z}''_k \leftarrow g(f_1(\mathbf{x}''^1_k), ..., f_n(\mathbf{x}''^n_k))$

        **for** $i \in [1..n]$ **do**

            $\mathbf{x}_k^i \leftarrow t_i(\mathbf{x}_k)$

            $\mathbf{z}_k^i \leftarrow g(f_i(\mathbf{x}_k^i))$

        **end for**

    **end for**

    **for** $i \in [1..n]$ **do**

$$\mathcal{L}_i \leftarrow -\frac{1}{2N} \left( \sum_{k=1}^{N} \log \frac{\exp \mathrm{sim}(\mathbf{z}_k^i, \mathbf{z}'_k)}{\sum_{l \neq k} \exp \mathrm{sim}(\mathbf{z}_k^i, \mathbf{z}'_l)} + \sum_{k=1}^{N} \log \frac{\exp \mathrm{sim}(\mathbf{z}_k^i, \mathbf{z}''_k)}{\sum_{l \neq k} \exp \mathrm{sim}(\mathbf{z}_k^i, \mathbf{z}''_l)} \right)$$

    **end for**

$$\mathcal{L} \leftarrow -\frac{1}{N} \sum_{k=1}^{N} \log \frac{\exp \mathrm{sim}(\mathbf{z}'_k, \mathbf{z}''_k)}{\sum_{l \neq k} \exp \mathrm{sim}(\mathbf{z}'_k, \mathbf{z}''_l)}$$

    $\mathcal{L}_{\text{CoMM}} \leftarrow \mathcal{L} + \sum_{i=1}^{n} \mathcal{L}_i$

    update $(f_i)_{i \in [1..n]}, g$ to minimize $\mathcal{L}_{\text{CoMM}}$

**end for**

**return** $(f_i)_{i \in [1..n]}, g$

---

Boston-area hospital between 2001 and 2012. It contains information about 53 423 hospital admissions, including 38 597 distinct patients. We use the data as provided by MultiBench (Liang et al., 2021), organized as in (Purushotham et al., 2018). There are two data modalities: first, a time series modality, composed by a set of medical measurements of a given patient taken every hour during 24 hours. Each measurement is a vector of size 12 (*i.e.*, including 12 different measured numerical values). Second, a static modality, including medical information about the patient (age, gender, *etc*.), represented as a vector of size 5 (tabular data). As in (Liang et al., 2023b), in our experiments we address the binary classification task of predicting whether a patient fits any disease in the ICD-9 code in group 7 (460-519). ICD-9 (*International Statistical Classification of Diseases and Related Health Problems*) codes are used to classify diseases and a variety of symptoms. Almost every health condition can be assigned a unique ICD-9 code group, where each group includes a set of similar diseases.

- **MOSI** (Zadeh et al., 2016) is a sentiment analysis dataset obtained from 2 199 YouTube video segments. Each sample consists of a video (visual frames), the corresponding audio and transcription (text). The original dataset evaluates sentiment intensities with continuous labels ranging from -3 to 3. We follow previous works (Liang et al., 2023b) and consider the binary version of the labels (positive and negative), and the same data modalities for training: text and videos.

- **UR-FUNNY** (Hasan et al., 2019) is a dataset for humor detection in human speech. It was created from 1 866 TED talk videos, obtaining more than 16 000 samples (parts of the videos). Each sample in the dataset consists of videos (visual frames), audio and their transcrips (text). The task is binary classification (humor or non-humor sequence).

- **MUsTARD** (Castro et al., 2019) is a multimodal video dataset for automated sarcasm detection. It contains videos from popular television shows including Friends, The golden girls, The big bang theory and Sarcasmaholics anonymous. Each sample in the dataset correspond to an utterance composed of a video (visual frames), its corresponding audio and the transcription (text), labeled as sarcastic or non-sarcastic. As previous works (Castro et al., 2019; Liang et al.,

2023b), we use the balanced partition consisting of 690 utterances. In our experiments with two modalities, we considered only text and vision.

- **Vision&Touch** (Lee et al., 2020) is a robot manipulation multimodal dataset including visual, force and proprioception data for a peg insertion task. The data is collected from a 7-DoF, torque-controlled Franka Panda robot, with a triangle peg attached to its end-effector. Its goal is to insert the peg into a triangle hole situated in a box attached to a table in front of the robot. By running a random policy (the robot takes random actions) and a heuristic policy (the robot attempts peg insertion), 150 trajectories are recorded, each of them consisting of 1 000 timesteps. These trajectories contain RGB images, depth maps, force, end-effector position and velocity. Following MultiBench (Liang et al., 2021), we consider two tasks on this dataset: (i) the binary task of predicting contact or no contact in the next timestep, and (ii) predicting the end-effector position (measured in MSE).

### E.3 MM-IMDb

Multimodal IMDb (MM-IMDb) (Arevalo et al., 2017) is a multimodal dataset for movie genre prediction. It has been built from the Movielens 20M dataset (GroupLens research, 2015) by filtering out movies without poster image. Therefore, MM-IMDb comprises 25 959 movies along with their plot, poster, genres and additional metadata (*e.g.* year, language, writer, *etc.*). As in previous works (Arevalo et al., 2017; Liang et al., 2021; Pérez-Rúa et al., 2019), we consider posters (image) and plots (text) as input data modalities to perform the multilabel classification genre prediction task (23 categories, each movie can be assigned to several categories). Technically, MM-IMDb is part of the Multibench benchmark (Liang et al., 2021), however, since we used raw data instead of the proposed pre-processed features, we treat it as a separate dataset.

## F    DETAILS ON PROCESSING TIMES AND MODEL COMPLEXITY

We include in Table 8 an analysis of the complexity of CoMM against CLIP and BLIP-2. As we can observe, the fusion module in CoMM adds a marginal computational cost to existing backbones without compromising speed.

| Model | FLOPs | MACs | #Params | Fwd-latency |
|---|---|---|---|---|
| CLIP | 251G | 126G | 222M | 488ms |
| CoMM (w/ CLIP) | 281G | 140G | 229M | 493ms |
| BLIP-2 | 9.22T | 4.61T | 1.17B | 14s |
| CoMM (w/ BLIP-2) | 9.48T | 4.74T | 1.17B | 15s |

Table 8: Comparison of model complexity and processing times of different multimodal architectures.

## G    PROOFS

**Proof 1 (Lemma 1)** *Indeed, combining Assumption 1 (multiview redundancy) and equations for $I(X_1; Y|X_2)$ and $I(X_2; Y|X_1)$ from Eq. (2) (main paper) we obtain:*

$$0 < I(X_1; Y|X_2) + I(X_2; Y|X_1) = U_1 + U_2 + 2S < 2\varepsilon_{info} \tag{8}$$

*Since $\varepsilon_{info}$ is supposed to be small and all the terms $U_1, U_2, S \geq 0$, their contributions are negligible.*

*Thus, under the multiview redundancy assumption $I(X_1, X_2; Y) \approx R$.* ∎

**Proof 2 (Lemma 2)** *Given data processing inequalities for the Markov chains $X \to X' \to Z'_\theta$ and $Z'_\theta \to X \to Z_\theta$, we have:*

$$I(Z_\theta; Z'_\theta) \leq I(X, Z'_\theta) \leq I(X, X') \tag{9}$$

*The equality can be achieved, for example, by selecting $f_\theta(\cdot) = Id(\cdot)$, the identity function.* ∎

**Proof 3 (Lemma 3)** *First, we prove that $I(Z'_{\theta^\star}; Y) = I(X', Y)$.*

*Indeed, we have:*

$$
\begin{aligned}
I(X'; Y) &= I(X'; Y; X) + I(X'; Y|X) \\
&= I(Z'_{\theta^\star}; Y; X) && \text{\textit{(by lemma 1 in (Wang et al., 2022a))}} \\
&= I(Z'_{\theta^\star}; Y) - I(Z'_{\theta^\star}; Y|X) \\
&= I(Z'_{\theta^\star}; Y) && \text{\textit{because } } Z'_{\theta^\star} = f_{\theta^\star}(t(X)) \quad (10)
\end{aligned}
$$

*Second, let $\mathcal{T} = \{t_i\}$ such that $X' = t_i(X) = X_i$ and $Z'_{\theta^\star} = f_{\theta^\star}(X_i) = Z_i$ for $i \in \{1, 2\}$ (with a slight abuse of notation). Thanks to the previous result (in Eq. (10)) and by the consistency equations for $I(X_i; Y)$ in Eq. (2) (main paper), the final result follows:*

$$
\begin{aligned}
I(Z_i; Y) &= I(Z'_{\theta^\star}; Y) \\
&= I(X'; Y) && \text{\textit{because of Eq. (10)}} \\
&= I(X_i; Y) \\
&= R + U_i && \text{\textit{because of consistency equations.}} \quad (11)
\end{aligned}
$$

∎

