# OpenReview forum: "What to align in multimodal contrastive learning?"
_ICLR.cc/2025/Conference — ICLR 2025 Poster_

### Official Review · Reviewer_gLzH · 2024-10-23

**Soundness:** 4
**Presentation:** 3
**Contribution:** 3
**Rating:** 8
**Confidence:** 3

**Summary:**

In this paper, the authors proposed CoMM, a novel multimodal contrastive representation learning method that performs multiple InfoNCE contrastive losses between unimodal representations and augmented multimodal representations in the same space. The authors first theoretically justified their approach through information theory and breaking down multimodal interactions into Redundancy, Uniqueness and Synergy. The authors evaluated their proposed method first on a synthetic dataset to show that CoMM can better capture the interactions, and then evaluated CoMM against other self-supervised representation learning methods on many datasets from MultiBench and MM-IMDB with linear evaluation classification accuracy, and CoMM outperformed all baselines. The authors also conducted ablation studies on the training objective and architecture of CoMM on the synthetic dataset.

**Strengths:**

1. The authors proposed a novel method for an important problem (self-supervised multimodal representation learning), CoMM, that outperformed all baselines in almost all tasks. CoMM also only adds a marginal computation overhead compared to baselines.

2. The authors theoretically justified why CoMM works, and confirmed their theory with experiments and analyses on the synthetic dataset.

3. The paper's presentation quality is generally good. No significant grammar errors and the figures/tables are well-made.

4. The authors justified their design choices through ablation studies on both synthetic and real datasets.

**Weaknesses:**

Minor presentation issue: In section 4.2, the title claims that all tasks in this subsection should be 2-modal, but in lines 356-358 some of the video-affect tasks are 3-modal. It is unclear which 2 modalities out of 3 were used for this experiment, and it can be quite confusing (especially on UR-funny which occurred in both the 2-modal experiments and the 3-modal experiments).

**Questions:**

(Not important, just out of curiosity) Why is MM-IMDB not listed as part of MultiBench? (Since MM-IMDB is included in MultiBench)

---

> ### Author Response · Authors · 2024-11-18
>
> Thank you for the accurate analysis of our work!
>
> ---
>
> > **Reviewer:** Minor presentation issue: In section 4.2, the title claims that all tasks in this subsection should be 2-modal, but in lines 356-358 some of the video-affect tasks are 3-modal. It is unclear which 2 modalities out of 3 were used for this experiment, and it can be quite confusing (especially on UR-funny which occurred in both the 2-modal experiments and the 3-modal experiments).
>
> Thank you for raising this point! We will slightly modify the section titles to "Experiments with $n$ modalities on real-world datasets" with $n=2,3$, which we believe are less prone to confusion. We decided to separate these two sections because Section 4.2 is fully comparable to the settings presented in FactorCL. Therefore, we used the same data modalities and backbone networks as FactorCL's authors, namely: vision (image frames) and text for MOSI, UR-FUNNY and MUsTARD; and images and proprioception data for Vision\&Touch. In section 4.3, we show that CoMM works with more than two modalities and that more modalities can improve the results, which is a current limitation of FactorCL.
>
> > **Reviewer:** (Not important, just out of curiosity) Why is MM-IMDB not listed as part of MultiBench? (Since MM-IMDB is included in MultiBench)
>
> Indeed, MM-IMDb is included in MultiBench, but we decided to treat it as a separate case because we used raw data in our experiments instead of the pre-processed features proposed by MultiBench. This is written in lines 1177-1179 in the Appendix.

---

> > ### Comment · Reviewer_gLzH · 2024-11-21
> > **Response**
> >
> > Thank you for your clarifications! My score remains positive.

---

### Official Review · Reviewer_qsrm · 2024-10-28

**Soundness:** 4
**Presentation:** 4
**Contribution:** 3
**Rating:** 8
**Confidence:** 4

**Summary:**

This work introduces a novel self-supervised learning objective for multimodal representation learning that captures the redundant, unique and synergetic interactions between different modalities. Their proposed approach (CoMM) is composed of a multimodal architecture, with modality-specific encoders and an attention-based fusion mechanism, that encodes a multimodal representation and a novel contrastive learning objective, grounded on the theory of partial information decomposition. The authors extensively evaluate CoMM against other self-supervised objectives, considering datasets of different complexities and number of modalities. The results highlight that the proposed method: (i) is able to capture the proposed interactions; (ii) outperforms other learning objectives in both self-supervised and fine-tuning settings; (iii) can be employed on scenarios with complex modalities (such as images and text) and with more than 2 modalities. Finally, the authors present an ablation study focused on the loss function, fusion module and data augmentation.

**Strengths:**

- **Originality**:
  - As highlighted by the authors, the use of self-supervision contrastive objectives to learn multimodal representations is not particularly novel (see, for example [1-4] for examples in fully-training and fine-tuning settings). However, the authors focus on the problem of learning not only redundant interactions between the modalities, but also the unique and synergetic interactions. This work seems to follow in line with previous work (FactorCL [6]), extending it to account for synergetic interactions and relaxing the factorization assumption and the need for the use of optimal multimodal augmentations. The use of the Trifeature dataset to quantitively evaluate the effects of redundant, unique and synergetic interactions is quite ingenious and, to the best of my knowledge, novel.

- **Quality**:
   - The work presented here is undoubtedly of high quality. The authors present in depth their main contribution, CoMM, and ground it in the theory of partial information decomposition. The theoretical results, even if not particularly extensive, give a strong support to the proposed method. Furthermore, the evaluation shown in this paper is also of high-quality: the authors evaluate their approach in both controlled and more complex environments, against a wide range of baselines, and the results mostly support the claims of the authors (see "Questions" for more details and experimental clarifications).

- **Clarity**:
   - The current version of the work is well-written, with no major typos (that I could detect). The authors employ a clear notation throughout the document. Furthermore, the document is easy to read, also due to the intuitive figures created by the authors (the only exception being Figure 1, where it is not clear what does $\mathcal{M}$ refers to). I would, however, recommend that the authors place the "Limitations and Future Work" section of the Appendix in the main document, as it clarifies the current limitations of the method (which is not something necessarly negative) for the reader.

- **Significance**:
   - This work proposes a novel multimodal learning objective that is able to capture a wide range of interactions between multimodal data for downstream tasks. Given the relevance of multimodal representation learning for the current field of AI, the work can have significant impact in both researchers and practitioners.


**References**:

- [1] Radford, Alec, et al. "Learning transferable visual models from natural language supervision." International conference on machine learning. PMLR, 2021.
- [2] Zhai, Xiaohua, et al. "Sigmoid loss for language image pre-training." Proceedings of the IEEE/CVF International Conference on Computer Vision. 2023.
- [3] Li, Junnan, et al. "Blip: Bootstrapping language-image pre-training for unified vision-language understanding and generation." International conference on machine learning. PMLR, 2022.
- [4] Poklukar, Petra, et al. "Geometric multimodal contrastive representation learning." International Conference on Machine Learning. PMLR, 2022.
- [5] Liang, Paul Pu, et al. "Factorized contrastive learning: Going beyond multi-view redundancy." Advances in Neural Information Processing Systems 36 (2024).

**Weaknesses:**

My two main concerns with the current version of the work are the following:
- From an architectural point-of-view, the work presents little novelty: the multimodal fusion mechanism, based on concatenation and a transformer block, is also not particularly novel (see, for example, [1], where the authors discuss several works that employ concatenation for multimodal inputs in transformer architectures). Also, despite introducing some novel ideas, the work is still a continuation of FactorCL [2], where the use of partial information decomposition theory was also explored to derive a suitable learning objective (even if a different one from the one proposed).
- The majority of evaluations presented by the authors are in controlled datasets (Section 4.1) or simplified environments (Section 4.2 - Multibench, Section 4.3), where the high-dimensional raw data are preprocessed into low-dimensional feature vectors (as described by the authors on Section 4.2). As such, it is not clear how the learned representations by the proposed method would work on more complex, real-world tasks (beyond the Multimodal IMDB dataset), for example on the ones proposed in [3].


**References**:

[1] - Xu, Peng, Xiatian Zhu, and David A. Clifton. "Multimodal learning with transformers: A survey." IEEE Transactions on Pattern Analysis and Machine Intelligence 45.10 (2023): 12113-12132.

[2] - Liang, Paul Pu, et al. "Factorized contrastive learning: Going beyond multi-view redundancy." Advances in Neural Information Processing Systems 36 (2024).

[3] - Ying, Kaining, et al. "MMT-Bench: A Comprehensive Multimodal Benchmark for Evaluating Large Vision-Language Models Towards Multitask AGI." Forty-first International Conference on Machine Learning.

**Questions:**

- **1**: The authors claim in the introduction that they evaluate CoMM across "diverse data types (images, text and audio)". However, from my understanding, the datasets used in the MultiBench benchmark are composed of pre-processed features extracted from the original, high-dimensional modalities. Am I correct? Given the use of pre-processed, low-dimensional, features, instead of raw data, it might be a stretch that claiming that the authors evaluate CoMM on audio data. Furthermore, how do you apply label-preserving multimodal transformations to these feature features?
- **2**: How to define the set of label-preserving multimodal transformations? Is it task-dependent? How large does this set need to be to work for your framework. Perhaps a section, even if in appendix, discussing these questions would be quite interesting, especially for practitioners.
- **3**: In Section 4.1., I found it interesting that there is no comparison with FactorCL, since this comparison would emphasize the relevance of the contribution (since, to my understanding, FactorCL cannot capture the synergetic interactions in the dataset). How does FactorCL perform in this controlled experiment?
- **4**: In Table 1, CoMM outperforms the other approaches across most datasets except for MIMIC. Any reason for this? Is there any special feature of the MIMIC dataset?
- **5**: In Section 5, Fusion module, did you also compare to a Concat+Non-linear fusion mechanism?

---

> ### Author Response · Authors · 2024-11-18
>
> Thank you very much for your careful and critical reading of our work!
>
>  ---
>
> >**Reviewer:** Despite introducing some novel ideas, the work is still a continuation of FactorCL, where the use of partial information decomposition theory was also explored to derive a suitable learning objective (even if a different one from the one proposed).
>
> We would like to highlight that the use of Partial Information Decomposition (PID) to derive contrastive learning objectives is, to the best of our knowledge, a novel contribution.
>     FactorCL uses traditional information theory (not PID) to derive its learning objectives, which limits their analysis to uniqueness and redundancy, **without integrating synergy**. This represents a crucial conceptual difference between FactorCL and our approach. By leveraging PID, we are able to derive a loss function that models **uniqueness, redundancy and synergy**.
>
> >**Reviewer:** The majority of evaluations presented by the authors are in controlled datasets (Section 4.1) or simplified environments (Section 4.2 - Multibench, Section 4.3), where the high-dimensional raw data are preprocessed into low-dimensional feature vectors (as described by the authors on Section 4.2). As such, it is not clear how the learned representations by the proposed method would work on more complex, real-world tasks (beyond the Multimodal IMDB dataset), for example on the ones proposed in MMTBench.
>
> We first would like to emphasize that CoMM was evaluated on 7 multimodal datasets, on 10 tasks, with a diversity of data domains and included experiments with two or three data modalities. Some of these datasets were indeed pre-processed and used as low-dimensional features for the sake of fair comparisons with competitor methods. For trifeatures, MM-IMDb and Vision\&Touch, we used raw features. We believe this proves how versatile CoMM is: we can use frozen encoders for any data modality (equivalent to using low-dimensional features) or train encoders from scratch, and the architecture of the encoders can be adapted to the data modality being considered.
>
> Finally, we agree that testing our representations in MMTBench is a great next step to study how to adapt the multimodal representations obtained through CoMM to generative tasks, but out of the scope of this work.
>
> >**Reviewer:** Given the use of pre-processed, low-dimensional, features, instead of raw data, it might be a stretch that claiming that the authors evaluate CoMM on audio data.
>
> Indeed, in some experiments we have used pre-processed features as proposed from MultiBench, because we needed our results to be comparable to those presented in FactorCL [1], which is the closest prior work. However, we have also included experiments with raw data in three datasets: Trifeatures, MM-IMDb, and Vision\&Touch. In particular, running the experiments on MM-IMDb on raw images and text was our own choice, in order to run comparisons with modern vision and language models like CLIP, BLIP-2 and LLaVA-NeXT.
>
> >**Reviewer:** Furthermore, how do you apply label-preserving multimodal transformations to these features?
>
> As detailed in Appendix Section B.2 (lines 940–953), the transformations applied to these pre-processed features—typically time-series data extracted from videos, health recordings, or force-torque readings—consist of a random composition of Gaussian noise and random sequence dropping, with the latter ranging from 0\% to 80\%.

---

> > ### Author Response · Authors · 2024-11-18
> >
> > >**Reviewer:** How to define the set of label-preserving multimodal transformations? Is it task-dependent? How large does this set need to be to work for your framework. Perhaps a section, even if in appendix, discussing these questions would be quite interesting, especially for practitioners.
> >
> > Thanks! Adding a section in the appendix is an excellent suggestion.
> >
> > Indeed, the set of optimal multimodal transformations is task-dependent. However, in practice, we have seen that the augmentations applied in our experiments are general enough to obtain good results in a variety of datasets (7 datasets, 10 tasks, diverse data modalities). These augmentations were chosen according to the findings in unimodal self-supervised learning [2, 3], and by a comprehensive ablation study on the best augmentation strategies applicable to time-series that can be found in the Appendix, Section C.1 (lines 1026 - 1040).
> >
> > Theoretically, the set of multimodal augmentations need to be large enough such that no information about the task $Y$ is lost, but small enough to extract this information only (Tian et al. [4] have referred to this as the _InfoMin principle_). In Section 5, lines 470-480, our data augmentation ablation study shows that if the set of augmentations is not large enough, then synergy cannot be learnt (the performance is always random chance). In order to further explore the feasibility of Assumption and inspired by the strategy developed by Tian et al., we evaluate CoMM by progressively increasing the strength of the data augmentations applied. We use random crop as the augmentation to control in the vision domain, mainly for two reasons: first, it is intuitive that by decreasing the level of cropping (keeping less information about the images), we are destroying the task-relevant information; and second, because it has been empirically demonstrated that cropping is a critical transformation in self-supervised learning for vision [2, 5]. For the text domain, we use masking as the augmentation to control.
> >
> > More specifically, on Trifeatures we randomly crop the original image in the first modality from 0% up to $x$% ($x=20$% is the strongest augmentation while $x=60$% is the lightest);  and from $x$% to 100% ($x=0.05$% is the strongest, $x=15$% the lightest).
> > For MM-IMDb, we also use random crop of the image modality from $x$% up to 100% and masking of text with a decreasing probability $x$% from 90% (the strongest) to 20% (the lightest).
> >
> > The results on the synergy task on the **bimodal trifeatures dataset** are in the following table:
> >
> > | Augmentation strength | $(0, 0.2)$ | $(0, 0.3)$ | $(0, 0.4)$ | $(0, 0.6)$ | $(0.05, 1)$ | $(0.1, 1)$ | $(0.12, 1)$ | $(0.15, 1)$ |
> > |:-----------------------|:-------:|:-------:|:-------:|:-------:|:-------:|:-------:|:-------:|:-------:|
> > | Accuracy              | 53.01 | 63.03 | 75.78 | 77.51 | 71.69  | 67.66  | 62.97  | 50.0  |
> >
> > And the results on **MM-IMDb**:
> >
> > | Augmentation strength | $C0.9,M0.9$ | $C0.9,M0.8$ | $C0.1,M0.7$ | $C0.1,M0.6$ | $C0.1,M0.5$ | $C0.2,M0.2$ | $C0.5,M0.2$ | $C0.7,M0.2$ | $C0.9,M0.2$ |
> > |:-----------------------|:-------:|:-------:|:-------:|:-------:|:-------:|:-------:|:-------:|:-------:|:-------:|
> > | Weighted F1           | 55.4  | 57.8  | 59.7  | 60.0  | 60.5  | 60.4  | 59.6  | 59.1  | 58.8  |
> >
> > Our results, both in the controlled environment of the bimodal Trifeatures dataset and in the real-world application of MM-IMDb, demonstrate that the sweet spot of the InfoMin principle can be reached. By gradually increasing the strength of the applied transformations, we enhance model performance by reducing noisy information, up to an optimal point (the sweet spot) where noise is minimized while task-relevant information is preserved. However, applying overly strong augmentations destroys task-relevant information, leading to a degradation in model performance.
> >
> > We have added a new section on the appendix entitled _On the feasibility of Assumption 1_ with these results, as the reviewer suggested. This analysis, can be a good starting points for practitioners to choose effective augmentations.
> >
> > In our experiments, we used the set of augmentations that was most suitable and general for all the considered tasks.

---

> > > ### Author Response · Authors · 2024-11-18
> > >
> > > >**Reviewer:**
> > > In Section 4.1., I found it interesting that there is no comparison with FactorCL, since this comparison would emphasize the relevance of the contribution (since, to my understanding, FactorCL cannot capture the synergetic interactions in the dataset). How does FactorCL perform in this controlled experiment?
> > >
> > >
> > > Indeed, FactorCL is an important comparison for our method. We ran the experiments, but the results are poor, especially for synergy (random chance). Here you can find these results:
> > >
> > > | **Model**         | **Redundancy**         | **Uniqueness**         | **Synergy**             | **Average**    |
> > > |:-------------------:|:----------------------:|:-----------------------:|:-----------------------:|:--------------:|
> > > | *Cross*    | $99.98_{\pm 0.02}$    | $11.62_{\pm 0.88}$     | $50.0_{\pm 0.0}$        |   $53.87$      |
> > > | **FactorCL**      | $99.82_{\pm 0.13}$    | $62.46_{\pm 2.60}$     | $46.50_{\pm 2.86}$  |   $69.56$    |
> > > | *Cross+Self*            | $99.73_{\pm 0.17}$| $86.86_{\pm 0.80}$ | $50.0_{\pm 0.04}$      |  $78.86$   |
> > > | **CoMM**            | **$99.92_{\pm 0.03}$**| **$87.83_{\pm 1.55}$** | **$71.87_{\pm 2.06}$**      | **$86.83$**    |
> > >
> > > We have added them to Fig. 4 for completeness.
> > >
> > >
> > > >**Reviewer:** In Table 1, CoMM outperforms the other approaches across most datasets except for MIMIC. Any reason for this? Is there any special feature of the MIMIC dataset?
> > >
> > > As shown by [6] (Table 3),  uniqueness ($U_1$) is the prevalent type of interaction related to the mortality prediction task for MIMIC, while the synergy between the two modalities is less relevant. This explains the difference of performance that we observe between CoMM and FactorCL in this dataset.
> > >
> > >
> > > >**Reviewer:** In Section 5, Fusion module, did you also compare to a Concat+Non-linear fusion mechanism?
> > >
> > > Thanks for the suggestion. We ran the experiments with Concat+Non-linear implemented as a 2-layered perceptron, the results are here:
> > >
> > > | **Fusion**         | **Redundancy**         | **Uniqueness**         | **Synergy**             | **Average**    |
> > > |:-------------------:|:----------------------:|:-----------------------:|:-----------------------:|:--------------:|
> > > | Concat + Linear     | $99.71_{\pm 0.06}$    | $81.49_{\pm 2.88}$     | $50.0_{\pm 0.0}$        | $77.07$        |
> > > | *Concat + MLP*      | $99.81_{\pm 0.04}$    | $83.37_{\pm 5.32}$     | **$76.33_{\pm 1.67}$**  | *$86.50$*      |
> > > | **CoMM**            | **$99.92_{\pm 0.03}$**| **$87.83_{\pm 1.55}$** | $71.87_{\pm 2.06}$      | **$86.83$**    |
> > >
> > >
> > > We observe that CoMM's fusion is the most suitable to learn all interactions (best average result). We also observe that the reviewer's suggestion (Concat+Non linear) is indeed very good at learning synergy in the trifeatures dataset.
> > >
> > > However, we believe that CoMM's fusion strategy is more versatile and better suited to any data modality --in particular for sequential data--,thanks to its transformer-based fusion. CoMM's versatility and efficiency have been demonstrated in our experiments with diverse data inputs. Concat+Non-linear is an interesting lead when using CNNs as encoders (as in the trifeature experiment).
> > >
> > > ### References
> > >
> > > [1] Paul Pu Liang, Zihao Deng, Martin Ma, James Zou, Louis-Philippe Morency, and Ruslan Salakhut-
> > > dinov. Factorized contrastive learning: Going beyond multi-view redundancy. Advances in Neural
> > > Information Processing Systems (NeurIPS), 36:32971–32998, 2023b.
> > >
> > > [2] Ting Chen, Simon Kornblith, Mohammad Norouzi, and Geoffrey Hinton. A simple framework for
> > > contrastive learning of visual representations. In International Conference on Machine Learning
> > > (ICML), pp. 1597–1607, 2020a.
> > >
> > > [3] Jacob Devlin, Ming-Wei Chang, Kenton Lee, and Kristina Toutanova. BERT: Pre-training of deep
> > > bidirectional transformers for language understanding. In Conference of the North American
> > > Chapter of the Association for Computational Linguistics: Human Language Technologies, Vol-
> > > ume 1 (Long and Short Papers), pp. 4171–4186, 2019.
> > >
> > > [4] Tian, Y., Sun, C., Poole, B., Krishnan, D., Schmid, C., & Isola, P. What makes
> > > for good views for contrastive learning?. Advances in neural information processing systems, 33,
> > > 6827-6839, 2020.
> > >
> > > [5] Kaiming He, Haoqi Fan, Yuxin Wu, Saining Xie, and Ross Girshick. Momentum contrast for
> > > unsupervised visual representation learning. In Proceedings of the IEEE/CVF Conference on
> > > Computer Vision and Pattern Recognition (CVPR), pp. 9729–9738, 2020.
> > >
> > > [6] Paul Pu Liang, Yun Cheng, Xiang Fan, Chun Kai Ling, Suzanne Nie, Richard Chen, Zihao Deng,
> > > Faisal Mahmood, Ruslan Salakhutdinov, and Louis-Philippe Morency. Quantifying & modeling
> > > feature interactions: An information decomposition framework. Advances in Neural Information
> > > Processing Systems (NeurIPS), 36:27351–27393, 2023a.

---

> > > > ### Comment · Reviewer_qsrm · 2024-11-21
> > > >
> > > > Thank you for the clarification on my questions. For now, I maintain my positive score as I believe the authors did a great job with this work.

---

### Official Review · Reviewer_ZXRV · 2024-11-04

**Soundness:** 2
**Presentation:** 2
**Contribution:** 2
**Rating:** 6
**Confidence:** 5

**Summary:**

This paper proposed CoMM, a Contrastive Multimodal learning framework, that captures multimodal interactions and aligns multimodal representations by maximizing the mutual information between augmented versions of these multimodal features. The paper shows that, under certain data augmentation, this framework can capture uniqueness and synergy better than existing work. Empirically, it is also shown that such framework facilitates multimodal learning on various multimodal benchmarks.

**Strengths:**

1. **Novel multimodal interaction learning under a contrastive learning framework**: the proposed CoMM framework is both theoretically and empirically shown to be able to capture uniqueness and (especially) synergy better than prior work.
2. **Comprehensive evaluation and careful ablation**: the proposed method is evaluated on a wide range of multimodal benchmarks, which involves diverse tasks, all requiring different level modeling of interactions. The paper conducts careful ablation in analyzing the importance of each part of the loss in capturing different interactions, which has provided insights in how those interactions are learnt with the proposed framework.

**Weaknesses:**

1. **Need for clarification about the Multi-view redundancy assumption**: the paper introduces the Multi-view redundancy assumption (Definition 1) to highlight the insufficiency of several existing works that propose cross-modal contrastive learning. Specifically, the paper shows that under this assumption, which states that "most task-relevant information is shared across modalities", cross-modal contrastive learning is primarily learning the redundancy interaction while ignoring the others interactions (uniqueness and synergy). The reviewer has the following questions:

    1. it is unclear how applicable this assumption is in the realistic settings: in fact, as many prior work estimating multimodal interaction in real-world multimodal datasets show, multimodal tasks can vary by a diverse range of combinations of those interactions;

    2. given this assumption, it seems that capturing redundancy would be sufficient already to learn task-relevant information, what makes it necessary to go beyond learning redundancy;

    3. do those existing contrastive learning frameworks still capture primarily redundancy, i.e. does Lemma 1 still hold, without the multi-view redundancy assumption?

2. **Strong assumption of Label-preserving multimodal augmentations (Assumption 1)**: the paper points out that existing methods like FactorCL requires strong and often unrealistic assumption about optimal multimodal augmentations based on conditional augmentations. Instead, the paper relaxes the assumption to label-preserving multimodal augmentations, which is still considered strong, as it is hard to exactly measure the mutual information between the original data and its augmented version $I(X,X')$, as well as between the data and its label $I(X,Y)$ ensuring $I(X,X')=I(X,Y)$ holds in a realistic setting. In fact, in the evaluation, the reviewer does not find any evidence to support that when evaluating on the real-world datasets, the augmentation indeed preserves the label. This not only makes the theory and the experiments detached, but also makes the choice of the "right" augmentation unclear, which is essential to learn the correct objective.
3. **Indirect evidence of CoMM's superior learning of synergy**: the paper highlights the novelty of the proposed framework, CoMM, in better capturing the synergistic interaction than existing methods. However, except Figure 4 (which does not include comparison with the most similar work FactorCL), which is shown on synthetic dataset, the paper does not provide any other direct evidence for showing the claimed superior learning of synergy when evaluating on real-world datasets. For example, a large margin of performance when evaluating on datasets with primary synergistic interaction (and minimal other types of interaction), for example, VQA, will constitute strong evidence to support the claim. Also, it'll be useful to provide estimates of the proportion of different types of interaction within each real-world datasets for evaluation to better understand the advantages of CoMM for modeling different combinations of interactions.
4. **Mysterious number of 50% accuracy for probing synergy**: except for the probing accuracy of synergy for CoMM being 71.87% (which gets round up to 71.9% in Figure 4), the probing accuracy of synergy for all other methods are all 50%. Does this number represent random chance? In other words, how is the probing accuracy of synergy evaluated? Is it only based on presence/absence? In Table 5, it is shown that without the crop augmentation for either modality 1 or modality 2, the synergy interaction seems not to be learnt. What is the intuition for the crop augmentation being the key to learning synergy? Overall, the reviewer feels an insufficient exploration and understanding of modeling synergy (the highlighted novelty of the paper) in the proposed framework, except for a little bit of discussion in the Ablation section (loss function), which only provides a hypothetical explanation.
5. **Missing details about the synthetic experiment setups**: the reviewer does not find documentation about how redundancy, uniqueness, and synergy are defined for the Trifeature dataset or how to control the amount of each type of interaction, in the main text or in the appendix. The paper should provide more details about the synthetic experiment setups, since the original Trifeature dataset is not designed for multimodal learning but rather for feature learning.

**Questions:**

Regard weakness 1, answering the following questions can help differentiate CoMM from existing contrastive methods and highlight the necessity for going beyond modeling redundancy:
1. How applicable is the Multi-view redundancy assumption in the realistic settings?
2. What makes it necessary to go beyond learning redundancy under the Multi-view redundancy assumption? Why are existing contrastive modeling approaches that focus on primary redundancy interaction insufficient?
3. Without the Multi-view redundancy assumption, i.e. for other tasks, do those existing contrastive learning framework still capture primarily redundancy?

Regard weakness 2, why is the Label-preserving multimodal augmentations (Assumption 1) considered a weaker and more realistic assumption? How can the mentioned mutual information quantity be measured in a realistic setting in order to choose the right augmentations?

Regard weakness 3, the reviewer expects to see more direct evidence that shows the superior modeling of synergistic interaction of CoMM compared to the existing strong methods (e.g. FactorCL) on real-world datasets.

Regard weakness 4, answering the following questions can provide additional insights into how the proposed framework better models synergy:
1. How should a 50% probing accuracy for synergy be interpreted? Is it only based on presence/absence?
2. What is the intuition for the crop augmentation being the key to learning synergy?

Regard weakness 5, please add more details on synthetic experiment setup as previously suggested.

---

> ### Author Response · Authors · 2024-11-18
>
> Thank you very much for the detailed and critical evaluation of our paper and the precise questions!
>
> ---
>
> ### Clarifications on the multiview redundancy assumption
>
> > **Reviewer:** How applicable is the Multi-view redundancy assumption in the realistic settings?
>
> The multi-view redundancy assumption can indeed be applicable to realistic settings. It is undeniable what cross-modalities contrastive methods, like CLIP [1] or ALIGN [2] --that primarily focus on extracting redundant information--, have achieved in zero-shot or few-shot transfer learning on visual classification tasks, by training on large-scale datasets of aligned images and captions.
>
> Yet, in this paper we argue that there are real-life applications where different modalities can provide unique information (not present in the others), or where modalities together can provide synergistic information (only present when modalities are simultaneously present). This is the case in healthcare with data coming from different medical sensors; in multimedia with data coming from videos, audio and text; robotics where multiple devices and instruments provide diverse data, just to name a few examples [3].
>
> Liang et al. [4] have recently conducted a human
> evaluation study over 30 multimodal benchmarks to determine the dominant interaction (redundancy, uniqueness or synergy). Out of the 30 listed datasets, less than half have redundancy as the primary interaction, showing the necessity of modeling uniqueness and synergy as well.
>
> In such applications, only modeling redundancy is insufficient to achieve optimal results.
>
> > **Reviewer:** What makes it necessary to go beyond learning redundancy under the Multi-view redundancy assumption?
>
> We fully agree with the reviewer on this point. If we know beforehand that the only interaction intervening for a certain application is redundancy (i.e., the multiview redundancy assumption holds), then there is no need to go beyond redundancy modeling.
>
> As we previously exposed, the multiview redundancy assumption is not satisfied in many applications. Moreover, our experiments (see Section 4.1) have shown that CoMM is able to successfully learn the three types of interactions --redundancy, uniqueness, and synergy--, without assuming multiview redundancy.
>
>
> > **Reviewer:** Why are existing contrastive modeling approaches that focus on primary redundancy interaction insufficient?
>
> Contrastive modeling approaches that focus on redundancy will not be able to capture uniqueness or synergy. Therefore, they will not achieve optimal results on tasks requiring unique or synergistic information.
>
> This is brought to light by our experiments on the bimodal trifeature dataset (see Section 4.1, Fig. 4 in lines 298 - 313), where we observe that _Cross_ (method focusing on redundancy) excels at the redundancy task (100\% accuracy), but fails at capturing uniqueness (11.6\% accuracy, only 1.6\% better than chance) and synergy (50\% accuracy, equal to chance).
>
> In contrast, CoMM obtains very good results on the redundancy and in the uniqueness tasks (99.9\% and 87.8\% accuracy, respectively), and is the only model able to capture synergy (71.9\% accuracy).
>
>
> > **Reviewer:** Without the Multi-view redundancy assumption, i.e. for other tasks, do those existing contrastive learning framework still capture primarily redundancy?
>
> Yes. This is shown by our experiments on the bimodal trifeature dataset (see Section 4.1, Fig. 4 in lines 298 - 313), and the answer to the previous question.

---

> ### Author Response · Authors · 2024-11-18
>
> ### Strong assumption of Label-preserving multimodal augmentation
>
> > **Reviewer:** why is the Label-preserving multimodal augmentations (Assumption 1) considered a weaker and more realistic assumption? How can the mentioned mutual information quantity be measured in a realistic setting in order to choose the right augmentations?
>
> We agree with the reviewer that Assumption 1 is still strong, but we argue that it is feasible in practice
>
> Indeed, it has been deeply studied in the self-supervised literature in computer vision [7, 8]. Tian et al. [7] established it as the _InfoMin Principle_: we should reduce the mutual information between views while keeping task-relevant information intact. In their work, Tian et al. also showed that data augmentation is an effective strategy to meet this InfoMin principle, by seeking stronger data augmentation to reduce mutual information until achieving a sweet spot.
>
> We started digging into this question in our ablation studies (see Section 5, Table 5, lines 470-480). Our results revealed that removing data augmentation on either modality (i.e. by keeping more information than the task-relevant one) prevents the model from learning synergy and its performance is also degraded for uniqueness and redundancy.
>
> To further explore the feasibility of Assumption 1 and inspired by the strategy developed by Tian et al., we evaluate CoMM by progressively increasing the strength of the data augmentations applied. In the image domain, we use random crop as the augmentation to control, mainly for two reasons: first, it is intuitive that by decreasing the level of cropping (keeping less information about the images), we are destroying the task-relevant information; and second, because it has been empirically demonstrated that cropping is a critical transformation in self-supervised learning for vision [9, 10]. For the text domain, we use masking as the augmentation to control.
>
> More specifically, on Trifeatures we randomly crop the original image in the first modality from 0% up to $x$% ($x=20$% is the strongest augmentation while $x=60$% is the lightest);  and from $x$% to 100% ($x=0.05$% is the strongest, $x=15$% the lightest).
> For MM-IMDb, we also use random crop of the image modality from $x$% up to 100% and masking of text with a decreasing probability $x$% from 90% (the strongest) to 20% (the lightest).
>
> The results on the synergy task on the **bimodal trifeatures dataset** are in the following table:
>
> | Augmentation strength | $(0, 0.2)$ | $(0, 0.3)$ | $(0, 0.4)$ | $(0, 0.6)$ | $(0.05, 1)$ | $(0.1, 1)$ | $(0.12, 1)$ | $(0.15, 1)$ |
> |:-----------------------|:-------:|:-------:|:-------:|:-------:|:-------:|:-------:|:-------:|:-------:|
> | Accuracy              | 53.01 | 63.03 | 75.78 | 77.51 | 71.69  | 67.66  | 62.97  | 50.0  |
>
> And the results on **MM-IMDb**:
>
> | Augmentation strength | $C0.9,M0.9$ | $C0.9,M0.8$ | $C0.1,M0.7$ | $C0.1,M0.6$ | $C0.1,M0.5$ | $C0.2,M0.2$ | $C0.5,M0.2$ | $C0.7,M0.2$ | $C0.9,M0.2$ |
> |:-----------------------|:-------:|:-------:|:-------:|:-------:|:-------:|:-------:|:-------:|:-------:|:-------:|
> | Weighted F1           | 55.4  | 57.8  | 59.7  | 60.0  | 60.5  | 60.4  | 59.6  | 59.1  | 58.8  |
>
> Our results, both in the controlled environment of the bimodal Trifeatures dataset and in the real-world application of MM-IMDb, **demonstrate that the sweet spot of the InfoMin principle can be reached**. By gradually increasing the strength of the applied transformations, we enhance model performance by reducing noisy information, up to an optimal point (the sweet spot) where noise is minimized while task-relevant information is preserved. However, applying overly strong augmentations destroys task-relevant information, leading to a degradation in model performance.
>
> Moreover, our empirical analysis (see Section 4) has shown that **CoMM, with a general set of data augmentations allows for state-of-the-art performance across numerous real-life scenarios**, including 7 datasets, 10 tasks and diverse data modalities.
>
> We have added a new section on the appendix entitled _On the feasibility of Assumption 1_ with these results. As the reviewer suggested, this analysis can be a good starting point for practitioners to choose effective augmentations.

---

> ### Author Response · Authors · 2024-11-18
>
> ### Indirect evidence of CoMM’s superior learning of synergy:
>
> > **Reviewer:** except Figure 4 (which does not include comparison with the most similar work FactorCL), which is shown on synthetic dataset, the paper does not provide any other direct evidence for showing the claimed superior learning of synergy when evaluating on real-world datasets. For example, a large margin of performance when evaluating on datasets with primary synergistic interaction (and minimal other types of interaction), for example, VQA, will constitute strong evidence to support the claim. Also, it'll be useful to provide estimates of the proportion of different types of interaction within each real-world datasets for evaluation to better understand the advantages of CoMM for modeling different combinations of interactions.
>
>
> We agree that FactorCL is an important baseline and one of the closest works to CoMM in multimodal representation learning by learning multimodal interactions beyond redundancy. Despite our efforts (a thorough grid search for hyperparameter tuning on learning rate --ranging from $10^{-5}$ to $10^{-2}$ -- and number of epochs --until 150 epochs), FactorCL does perform poorly especially on synergy.
> This new baseline has been added in Figure 4 for completeness.
>
> | **Model**         | **Redundancy**         | **Uniqueness**         | **Synergy**             | **Average**    |
> |:-------------------:|:----------------------:|:-----------------------:|:-----------------------:|:--------------:|
> | *Cross*    | $99.98_{\pm 0.02}$    | $11.62_{\pm 0.88}$     | $50.0_{\pm 0.0}$        |   $53.87$      |
> | **FactorCL**      | $99.82_{\pm 0.13}$    | $62.46_{\pm 2.60}$     | $46.50_{\pm 2.86}$  |   $69.56$    |
> | *Cross+Self*            | $99.73_{\pm 0.17}$| $86.86_{\pm 0.80}$ | $50.0_{\pm 0.04}$      |  $78.86$   |
> | **CoMM**            | **$99.92_{\pm 0.03}$**| **$87.83_{\pm 1.55}$** | **$71.87_{\pm 2.06}$**      | **$86.83$**    |
>
>
> Measuring multimodal interactions as defined by PID is challenging  since it involves estimating information-theoretic measures [5]. The problem is even harder for high-dimensional data distributions. Liang et al. [5] have provided estimators for PID, and according to these measures, synergy is the dominant interaction in UR-FUNNY and MUsTARD, which is consistent with our results (see Table 1) where CoMM ourperforms FactorCL by 2.6\% and 8.1\%, respectively.
>
> Very recently, Liang et al. [4] (paper to appear in NeurIPS Datasets and Benchmarks 2024), conducted a human evaluation study over 30 multimodal datasets to determine the dominant interaction.
> According to this study, for many VQA tasks, redundancy—not synergy—is the predominant interaction. Moreover, the multilabel classification problem MM-IMDb was found to be dominated by synergy, where again CoMM outperforms the second best method by 7.5\% and 5.8\% on macro and weighted F1-scores, respectively (see Table 2).

---

> ### Author Response · Authors · 2024-11-18
>
> ### Missing details about the synthetic experiment setups
>
> > **Reviewer:** Please add more details on synthetic experiment setup as previously suggested.
>
> We apologize that this experiment was not clear enough. Indeed, the original Trifeature dataset [6] was proposed as a controlled environment to study how vision neural networks learn different features (texture, shape and color). Since quantifying multimodal interactions in high-dimensional continuous data is a hard problem [5], we decided to leverage the properties of Trifeature to design a controlled environment (yet not too simple) to test the models' capacity to learn different multimodal interactions.
>
> As explained in the Appendix, Section E.1 (lines 1109 - 1119), the original trifeature dataset is composed of images containing one of ten shapes, rendered in one of ten textures, and displayed in one of ten colors. In total, there are 1\,000 combinations of these 3 features. To generate these images shapes are rendered within a 128 $\times$ 128 square rotated at an angle drawn between $[-45^{\circ}, 45^{\circ}]$ and placed at a random position within a larger image ($224 \times 224$), such that the shape is fully contained in the image. Then, an independently rotated texture and a color are applied.
>
> To generate our trifeature dataset, we considered the 1\,000 combinations of the three features and split them into 800 combinations for training and 200 for evaluation. To have more variety in the training set for training, each combination was generated 3 times (the shape and the texture were randomly rotated), obtaining a training split of 2\,400 images.
>
> The bimodal Trifeature dataset used in our experiments was built by considering the trifeature dataset twice (as two separate modalities) and building pairs from these two dataset copies. In total, we get 5\,760\,000 pairs (2\,400 $\times$ 2\,400) available for training, and 40\,000 (200 $\times 200$) available for evaluation.
>
> To create a controlled environment for evaluation of multimodal interaction learning, we needed to carefully design tasks where the dominant interaction was clearly defined.
>
> 1. To measure uniqueness $U_1$ (resp. $U_2$), given a pair of trifeature images, the task is to predict the texture of the first (resp. the second) image. The task is then a 10-class classification problem and chance level is at 10\%.
>
> 2. To measure redundancy $R$, given a pair of trifeature images with the same shape (but different color and texture), the task is to predict the shape of the pair (therefore, the redundant information that can be extracted either from the first or the second image). The task is then a 10-class classification problem and chance level is at 10\%.
>
> 3. To measure synergy $S$, the definition of the task was more subtle as it should require information from both modalities simultaneously and should not be possible to perform it from one of the images alone. To achieve this, we defined a mapping $\mathcal{M}$ between the ten textures and the ten colors (e.g. stripes=red, dots=green, etc.). Then, given a pair of trifeature images, the task is to predict whether the pair satisfies the mapping or not. The task is then a binary classification problem and chance level is at 50\%.
>
> To evaluate these tasks, we built two versions of the bimodal trifeature dataset:
>
> 1. For uniqueness and redundancy, we considered 10\,000 image pairs (out of the 5\,760\,000 pairs) for training and 4\,096 for testing, that have the same shape (to measure redundancy) and different texture (to measure uniqueness).
>
> 2. For synergy, we considered 10\,000 image pairs that respect the mapping $\mathcal{M}$ and used the same test set as before (4\,096 image pairs).
>
> Following your suggestion we added a new section in the appendix (Section B.5) entitled _Experimental settings in on the bimodal trifeature dataset_, which provides details on this experimetal design.
>
> #### Mysterious number of 50% accuracy for probing synergy
>
> > **Reviewer:** How should a 50\% probing accuracy for synergy be interpreted?
>
> Again, we apologize this was not clear enough in the original submission. As explained in the previous reply, **yes, 50\% is the random chance for the task evaluating synergy**. This task was cast as a binary classification problem of a pair of images belonging to a mapping $\mathcal{M}$ or not. Random chance was subtly mentioned in line 342, we have added it to Fig. 4 for more clarity. In section B.5 that we have added to the appendix we detail the evaluation setting for synergy (as in the previous reply).

---

> ### Author Response · Authors · 2024-11-18
>
> > **Reviewer:**
> What is the intuition for the crop augmentation being the key to learning synergy?
>
> The intuition for the crop augmentation being key to learning synergy in the trifeature dataset can be found in Table 5. **It is because cropping is crucial to learn the texture of the images**, and texture is directly related to the uniqueness tasks and the synergy task: $U_1$ is measured by predicting the texture of the first image, $U_2$ is measured by predicting the texture of the second image, and $S$ is measured by the mapping defined between the texture of the first image and the color of the second one.
>
> Indeed, by looking at lines 3 and 4 of Table 5, we observe that when we remove crop augmentation for the first modality, the model does not learn correctly $U_1$ (i.e. texture for the first modality), and similarly, when we remove crop augmentation for the second modality, the model does not learn correctly $U_2$.
>
>
> ### References
>
> [1] Alec Radford, Jong Wook Kim, Chris Hallacy, Aditya Ramesh, Gabriel Goh, Sandhini Agar-
> wal, Girish Sastry, Amanda Askell, Pamela Mishkin, Jack Clark, Gretchen Krueger, and Ilya
> Sutskever. Learning transferable visual models from natural language supervision. In International
> Conference on Machine Learning (ICML), pp. 8748–8763, 2021.
>
> [2] Chao Jia, Yinfei Yang, Ye Xia, Yi-Ting Chen, Zarana Parekh, Hieu Pham, Quoc Le, Yun-Hsuan
> Sung, Zhen Li, and Tom Duerig. Scaling up visual and vision-language representation learning
> with noisy text supervision. In International Conference on Machine Learning (ICML), pp. 4904–
> 4916, 2021.
>
> [3] Paul Pu Liang, Yiwei Lyu, Xiang Fan, Zetian Wu, Yun Cheng, Jason Wu, Leslie Chen, Peter Wu,
> Michelle A Lee, Yuke Zhu, et al. MultiBench: Multiscale benchmarks for multimodal represen-
> tation learning. In Neural Information Processing Systems (NeurIPS) – Track on Datasets and
> Benchmarks, volume 1, 2021.
>
> [4] Liang, P. P., Goindani, A., Chafekar, T., Mathur, L., Yu, H., Salakhutdinov, R., & Morency,
> L. P. (2024). Hemm: Holistic evaluation of multimodal foundation models. arXiv preprint arXiv:2407.03418.
>
> [5] Paul Pu Liang, Yun Cheng, Xiang Fan, Chun Kai Ling, Suzanne Nie, Richard Chen, Zihao Deng,
> Faisal Mahmood, Ruslan Salakhutdinov, and Louis-Philippe Morency. Quantifying & modeling
> feature interactions: An information decomposition framework. Advances in Neural Information
> Processing Systems (NeurIPS), 36:27351–27393, 2023a.
>
> [6] Katherine Hermann and Andrew Lampinen. What shapes feature representations? exploring
> datasets, architectures, and training. Advances in Neural Information Processing Systems
> (NeurIPS), 33:9995–10006, 2020.
>
> [7] Y. Tian, C. Sun, B. Poole, D. Krishnan, C. Schmid, and P. Isola, “What makes for good views
> for contrastive learning?,” Advances in neural information processing systems (NeurIPS), vol. 33,
> pp. 6827–6839, 2020.
>
> [8] R. Shwartz-Ziv and Y. LeCun, “To compress or not to compress–self-supervised learning and infor-
> mation theory: A review,” arXiv:2304.09355, 2023.
>
> [9] T. Chen, S. Kornblith, M. Norouzi, and G. Hinton, “A simple framework for contrastive learning of
> visual representations,” in International Conference on Machine Learning (ICML), pp. 1597–1607,
> 2020.
>
> [10] K. He, H. Fan, Y. Wu, S. Xie, and R. Girshick, “Momentum contrast for unsupervised visual repre-
> sentation learning,” in Proceedings of the IEEE/CVF Conference on Computer Vision and Pattern
> Recognition (CVPR), pp. 9729–9738, 2020.

---

> ### Author Response · Authors · 2024-11-21
> **Friendly reminder: Rebuttal period for ICLR submission**
>
> Dear Reviewer ZXRV,
>
> We are writing to kindly remind you that the rebuttal phase for ICLR submission is ongoing. We posted our responses and paper revision three days ago.
>
> Regarding weakness 2, we have provided empirical evidence to support Assumption 1, both through experiments conducted in a controlled environment and by analyzing a real-life dataset, as detailed in Section C.4 _On the feasibility of Assumption 1_. Additionally, our findings are corroborated by relevant literature.
>
> Regarding other weaknesses, we have provided further clarifications on our approach and replied to your questions.
>
> **If you have any additional feedback, concerns, or questions regarding our response, we would greatly appreciate hearing from you and would welcome further discussion.**

---

> > ### Comment · Reviewer_ZXRV · 2024-11-21
> >
> > Thanks for the author's responses and efforts in the additional experiments. I think most of my concerns are properly addressed. I'm willing to raising my rating to 6. After incorporating the rebuttal into the main paper, I believe the work now can contribute to the body of work that focuses diverse types of interactions for better multimodal modeling.

---

### Official Review · Reviewer_J1jX · 2024-11-09

**Soundness:** 1
**Presentation:** 2
**Contribution:** 2
**Rating:** 3
**Confidence:** 4

**Summary:**

The authors present a multimodal representation learning method that uses multimodal augmentations to learn all four multimodal interaction terms. The proposed optimization terms are designed to optimize for $R + U_i$ and $U_1 + U_2 + R + S$. The authors design a multimodal encoder+fusion architecture that achieve state-of-the-art performance on several MM benchmarks.

**Strengths:**

This paper grounds multimodal representation learning in the theoretical framework of multimodal information theory. Lemma 2 & 3 offer insight into novel methods that can be applied to learn multimodal representations with beyond redundancy.

**Weaknesses:**

I have strong doubts about the soundness of the theoretical foundation of this paper:
1. Assumption 1 assumes the existence of a multimodal augmentation $t$ such that $I(X; t(X)) = I(X; Y)$. This means that if $X$ contains more information than the label $Y$, $t(X)$ has to "reduce" the information in $X$ down to $Y$. This is different from just "label-preserving multimodal augmentation". The experiments section also offers little insight about how exactly this augmentation is carried out. The augmentations mentioned for real-world datasets, including cropping, jitter, etc., looks like they certainly would lead to $I(X; t(X)) \gg I(X; Y)$. This would be a direct contradiction of Assumption 1.
2. Moreover, for the pretraining paradigm used on real-world datasets, we don't even know at pretraining time what the task $Y$ is yet. Correct me if I'm wrong--it's not clear to me how it is possible for this assumption to hold.
3. The real gains from this method seems to be the aggressive mutual information optimization between all pairs of single modalities and all modalities as well as all data augmentations.

It would be helpful if the authors could provide more rigorous justification for Assumption 1. For example, why does Assumption 1 hold in the real world?

**Questions:**

See Weaknesses for questions about theoretical foundation of this paper.

In Lemma 2, it is shown that optimum is achieved assuming "an expressive enough network". Do we have any idea how big/complex the network needs to be to become an expressive enough network? How does performance scale with your network's size, etc.?

---

> ### Author Response · Authors · 2024-11-18
>
> Thank you for  your interesting feedback.
>
> ---
>
> > **Reviewer:** Assumption 1 assumes the existence of a multimodal augmentation $t$ such that $I(X;t(X)) = I(X;Y)$. This means that if $X$  contains more information than the label $Y$,  has to "reduce" the information in $X$ down to $Y$. This is different from just "label-preserving multimodal augmentation".
>
> Thank you for pointing this out. You are right, Assumption 1 is more than _just_ label-preserving augmentations, and its current name is misleading. For this reason we have changed the name to _Minimal label-preserving multimodal augmentations_.
>
>
> > **Reviewer:** The augmentations mentioned for real-world datasets, including cropping, jitter, etc., looks like they certainly would lead to $I(X;t(X)) \geq I(X;Y)$. This would be a direct contradiction of Assumption 1. Why does Assumption 1 hold in the real world?
>
> This hypothesis has been deeply studied in the self-supervised literature in computer vision [1, 2]. Tian et al. [1] established it as the _**InfoMin Principle**_: we should reduce the mutual information between views while keeping task-relevant information intact. In their work, Tian et al.  also showed that data augmentation is an effective strategy to meet this InfoMin principle, by seeking stronger data augmentation to reduce mutual information towards a sweet spot.
>
> We agree with the reviewer that if $I(X;t(X))$ is much bigger than $I(X;Y)$, then the assumption would be violated and the model would be unable to learn multimodal interactions (especially synergy). However, **we argue that the set of augmentations used throughout our experiments allow for the assumption to be approximately satisfied**.
>
> We started digging into this question in our ablation studies (see Section 5, Table 5, lines 470-480). Our results reveal that removing data augmentation on either modality (i.e. by keeping more information than the task-relevant one) prevents the model from learning synergy and its performance is also degraded for uniqueness and redundancy.
>
> To further explore the feasibility of Assumption 1 and inspired by the strategy developed by Tian et al., we evaluate CoMM by increasing the strength of the data augmentations applied. In the image domain, we use random crop as the augmentation to control for two reasons: (1) it is intuitive that by decreasing the level of cropping (keeping less information about the images), we are destroying information -- that will be, at some point, task-relevant; and (2) because it has been empirically demonstrated that cropping is a critical transformation in self-supervised learning for vision [3, 4]. For the text domain, we use masking as the augmentation to control.
> More specifically, on Trifeatures we randomly crop the original image in the first modality from 0% up to $x$% ($x=20$% is the strongest augmentation while $x=60$% is the lightest);  and from $x$% to 100% ($x=0.05$% is the strongest, $x=15$% the lightest).
> For MM-IMDb, we also use random crop of the image modality from $x$% up to 100% and masking of text with a decreasing probability $x$% from 90% (the strongest) to 20% (the lightest).
>
> The results on the synergy task on the **bimodal trifeatures dataset** are in the following table:
>
> | Augmentation strength | $(0, 0.2)$ | $(0, 0.3)$ | $(0, 0.4)$ | $(0, 0.6)$ | $(0.05, 1)$ | $(0.1, 1)$ | $(0.12, 1)$ | $(0.15, 1)$ |
> |:-----------------------|:-------:|:-------:|:-------:|:-------:|:-------:|:-------:|:-------:|:-------:|
> | Accuracy              | 53.01 | 63.03 | 75.78 | 77.51 | 71.69  | 67.66  | 62.97  | 50.0  |
>
> And the results on the **MM-IMDb**:
>
> | Augmentation strength | $C0.9,M0.9$ | $C0.9,M0.8$ | $C0.1,M0.7$ | $C0.1,M0.6$ | $C0.1,M0.5$ | $C0.2,M0.2$ | $C0.5,M0.2$ | $C0.7,M0.2$ | $C0.9,M0.2$ |
> |:-----------------------|:-------:|:-------:|:-------:|:-------:|:-------:|:-------:|:-------:|:-------:|:-------:|
> | Weighted F1           | 55.4  | 57.8  | 59.7  | 60.0  | 60.5  | 60.4  | 59.6  | 59.1  | 58.8  |
>
> Our results, both in the controlled environment of the bimodal Trifeatures dataset and in the real-world application of MM-IMDb, **demonstrate that the sweet spot of the InfoMin principle can be reached**. By gradually increasing the strength of the applied transformations, we enhance model performance by reducing noisy information, up to an optimal point (the sweet spot) where noise is minimized while task-relevant information is preserved. However, applying overly strong augmentations destroys task-relevant information, leading to a degradation in model performance.
>
> Moreover, our empirical analysis (see Section 4) has shown that **CoMM, with a general set of data augmentations allows for state-of-the-art performance across numerous real-life scenarios,** including 7 datasets, 10 tasks and diverse data modalities.
>
> We have added a new section on the appendix entitled _On the feasibility of Assumption 1_ with these results.

---

> > ### Author Response · Authors · 2024-11-18
> >
> > > **Reviewer:** The experiments section also offers little insight about how exactly this augmentation is carried out.
> >
> > We apologize that this is not explicit in the main paper. However, for the sake of reproducibility and open science, we have provided all details about our experimental settings and implementations in the Appendix. In particular, section B.2 _Data augmentation by modality_ (lines 940 - 953) precises the data augmentations applied to each modality.
> >
> >
> >
> > > **Reviewer:** Moreover, for the pretraining paradigm used on real-world datasets, we don't even know at pretraining time what the task  is yet. Correct me if I'm wrong--it's not clear to me how it is possible for this assumption to hold.
> >
> > It is true that the set of optimal augmentations is, in theory, task-dependent. However, there is ample evidence from the contrastive learning literature in the vision domain (unimodal) [3,4], showing that a well-chosen, unique set of augmentations enables the learning of general representations that achieve remarkable performance accross a wide range of vision tasks.
> >
> > Similarly, in our experiments (see Section 5) across 7 datasets—Trifeatures (3 tasks), MIMIC, MOSI, UR-FUNNY, MUsTARD, Vision\&Touch (2 tasks), and MM-IMDb—covering a total of 10 multimodal tasks with different data modalities, CoMM outperforms all competing methods (except for one task). This demonstrates that a single set of generic augmentations can be effective accross many multimodal tasks.
> >
> > Finally, since the optimal set of multimodal augmentations is task-dependent, knowing the task beforehand would allow for customizing the set of augmentations, potentially leading to better performance. This is supported by the experiments presented in the appendix, section C.1 (lines 1026 - 1040) and Fig. 6 (lines 972 - 997), where we show that by applying custom augmentations (multi-crop and multi-drop) to time-series data, CoMM can obtain even better results than by applying the generic set.
> >
> >
> >
> > > **Reviewer:**
> > The real gains from this method seems to be the aggressive mutual information optimization between all pairs of single modalities and all modalities as well as all data augmentations.
> >
> > We respectfully disagree with the reviewer's statement. As thoroughly demonstrated in our experiments on 7 datasets: trifeatures (3 tasks), MIMIC, MOSI, UR-FUNNY, MUsTARD, Vison\&Touch (2 tasks) and MM-IMDb; and a total of 10 multimodal tasks. CoMM is able to learn multimodal interactions and outperforms competitor methods in all (except one) tasks.
> >
> > Our study in a carefully designed controlled environment (bimodal trifeature dataset) (in Section 4.1) shows that:
> >
> > 1. only optimizing mutual information between modalities (cross methods) allows to learn redundant information, but this strategy is not enough to learn uniqueness nor synergy.
> >
> > 2. optimizing mutual information between modalities and inter-modalities (cross+self methods) allows to learn redundancy and uniqueness, but cannot capture synergy.
> >
> > 3. CoMM is the only method able to learn the three types of interactions.
> >
> > More specifically, CoMM is able to capture these interactions thanks to the combination of two loss terms: $\mathcal{L}_{\text{CoMM}} = \mathcal{L} + \sum_i \mathcal{L}_i$ (see Eq. (7)). Our ablation studies in Section 5 (lines 452-458) and Fig. 5 (lines 432-445) show that they are both important for the success of our method:
> > 1. As supported by Lemma 3, $\sum_i \mathcal{L}_i$ learns redundancy and uniqueness, but fails for synergy.
> >
> > 2. As supported by Lemma 2, $\mathcal{L}$ allows to learn all the interaction terms, but very slowly.
> >
> > 3. The combination of both enables an efficient learning of all interaction terms.
> >
> > These findings are further supported by the results of the ablation study about loss function on the MM-IMDb dataset, where the combination of the two losses (CoMM) achieves the best performance in this dataset known for needing the synergy between the two modalities [5] to perform the multi-label classification task.

---

> > > ### Author Response · Authors · 2024-11-18
> > >
> > > > **Reviewer:**
> > > In Lemma 2, it is shown that optimum is achieved assuming "an expressive enough network". Do we have any idea how big/complex the network needs to be to become an expressive enough network? How does performance scale with your network's size, etc.?
> > >
> > > Yes, the size of the backbones has an impact on the results and the best backbones might depend on the dataset and data modalities. In the Appendix, Section C.3 (lines 1057-1072), we performed ablation studies on the backbone size on Trifeatures dataset, comparing AlexNet with ResNet-50.
> > >
> > >
> > > | Model   | Redundancy       | Uniqueness     | Synergy        |
> > > |:---------:|:------------------------:|:----------------------:|:---------------------:|
> > > | AlexNet | $99.9_{\pm 0.03}$      | $87.8_{\pm 1.6}$     | $71.9_{\pm 2.0}$    |
> > > | ResNet  | $99.9_{\pm 0.03}$      | $96.3_{\pm 1.3}$     | $75.0_{\pm 1.7}$    |
> > >
> > >
> > > CoMM's results are better with the biggest backbone. Similarly, the experiments on MM-IMDb dataset (Table 2, lines 378 - 394) also show that CoMM performs better with bigger backbones: 61.48\% with CLIP (smaller backbone), against 64.75\% of weighted-F1 with BLIP-2 (bigger backbone). In our experiments on MultiBench datasets, we chose the backbones that have shown the best performance and that have been used by competitor methods like FactorCL [6, 7].
> > >
> > > Finally, we recall that by the Universal Approximation Theorem,
> > > _"standard multilayer feedforward networks with as few as one hidden layer using arbitrary squashing functions are capable of approximating any Borel measurable function from one finite dimensional space to another to any desired degree of accuracy, provided sufficiently many hidden units are available"_ [8].
> > >
> > > ### References
> > >
> > > [1] Y. Tian, C. Sun, B. Poole, D. Krishnan, C. Schmid, and P. Isola, “What makes for good views
> > > for contrastive learning?,” Advances in neural information processing systems (NeurIPS), vol. 33,
> > > pp. 6827–6839, 2020.
> > >
> > > [2] R. Shwartz-Ziv and Y. LeCun, “To compress or not to compress–self-supervised learning and infor-
> > > mation theory: A review,” arXiv:2304.09355, 2023.
> > >
> > > [3] T. Chen, S. Kornblith, M. Norouzi, and G. Hinton, “A simple framework for contrastive learning of
> > > visual representations,” in International Conference on Machine Learning (ICML), pp. 1597–1607,
> > > 2020.
> > >
> > > [4] K. He, H. Fan, Y. Wu, S. Xie, and R. Girshick, “Momentum contrast for unsupervised visual repre-
> > > sentation learning,” in Proceedings of the IEEE/CVF Conference on Computer Vision and Pattern
> > > Recognition (CVPR), pp. 9729–9738, 2020.
> > >
> > > [5] P. P. Liang, A. Goindani, T. Chafekar, L. Mathur, H. Yu, R. Salakhutdinov, and L.-P. Morency,
> > > “HEMM: Holistic evaluation of multimodal foundation models,” arXiv preprint arXiv:2407.03418,
> > > 2024.
> > >
> > > [6] P. P. Liang, Y. Lyu, X. Fan, Z. Wu, Y. Cheng, J. Wu, L. Chen, P. Wu, M. A. Lee, Y. Zhu, et al.,
> > > “MultiBench: Multiscale benchmarks for multimodal representation learning,” in Neural Information
> > > Processing Systems (NeurIPS) – Track on Datasets and Benchmarks, vol. 1, 2021.
> > >
> > > [7] P. P. Liang, Z. Deng, M. Ma, J. Zou, L.-P. Morency, and R. Salakhutdinov, “Factorized contrastive
> > > learning: Going beyond multi-view redundancy,” Advances in Neural Information Processing Systems
> > > (NeurIPS), vol. 36, pp. 32971–32998, 2023.
> > >
> > > [8] K. Hornik, M. Stinchcombe, and H. White, “Multilayer feedforward networks are universal approxi-
> > > mators,” Neural networks, vol. 2, no. 5, pp. 359–366, 1989.

---

> > > > ### Author Response · Authors · 2024-11-21
> > > > **Friendly reminder: Rebuttal period for ICLR submission**
> > > >
> > > > Dear Reviewer J1jX,
> > > >
> > > > We are writing to kindly remind you that the rebuttal phase for ICLR submission is ongoing. We posted our responses and paper revision three days ago.
> > > >
> > > > We have provided empirical evidence to support Assumption 1, both through experiments conducted in a controlled environment and by analyzing a real-life dataset, as detailed in Section C.4 _On the feasibility of Assumption 1_. Additionally, our findings are corroborated by relevant literature.
> > > >
> > > > **If you have any additional feedback, concerns, or questions regarding our response, we would greatly appreciate hearing from you and would welcome further discussion.**
> > > >
> > > > Best regards,
> > > > Authors.

---

### Author Response · Authors · 2024-12-03
**General response to reviewers**

We would like to thank the reviewers for their thoughtful feedback and suggestions.
Overall, the reviewers highlighted the following strengths of our paper:
1. **Originality**: Novel multimodal interaction learning under a contrastive learning framework; the proposed CoMM framework is both theoretically and empirically shown to be able to capture uniqueness and (especially) synergy better than prior work (R-ZXRV, R-qsrm, R-gLzH). On the empirical side, the experiments on the trifeature dataset are ingenious and novel (R-qsrm).

2. **Quality**:  The theoretical results give a strong support to the proposed method (R-qsrm, R-gLzH). The empirical evaluation shown in this paper is also of high-quality: the authors evaluate their approach in both controlled and more complex environments, against a wide range of baselines (R-ZXRV, R-qsrm), outperforming all baselines in almost all tasks while adding only a marginal computation overhead (R-gLzH). The authors justified their design choices through ablation studies on both synthetic and real datasets (R-ZXRV, R-gLzH).

3. **Significance**: This work proposes a novel multimodal learning objective that is able to capture a wide range of interactions between multimodal data for downstream tasks. Given the relevance of multimodal representation learning for the current field of AI, the work can have significant impact in both researchers and practitioners (R-ZXRV, R-qsrm).
On the theoretical side, Lemma 2 & 3 offer insight into novel methods that can be applied to learn multimodal representations (R-J1jX).

4. **Clarity**: The work is well-written and easy to read, also due to intuitive figures and tables (R-qsrm, R-gLzH).

The reviewers have provided valuable insights and identified weaknesses in our work, which we have addressed thoroughly. Specifically, we clarified a common assumption in multimodal contrastive learning (i.e., multiview redundancy), detailed our experimental settings, and demonstrated CoMM's ability to capture synergy in real-world scenarios (R-ZXRV). Additionally, we highlighted the critical conceptual differences between CoMM and prior work (R-qsrm).

One major concern (raised by R-J1jX and R-ZXRV) was related to the **feasibility of Assumption 1**. In response, **we have provided robust evidence supporting this assumption**, leveraging an experimental protocol initially designed for the unimodal case that we have extended to the multimodal setting. We have added a new section in the Appendix to present and discuss these results.

**We believe that all these constructive comments have contributed to increase the quality of this work.**

---

### Meta-Review · Area_Chair_SqwZ · 2024-12-20

**Metareview:**

This paper proposes CoMM, a Contrastive Multimodal learning framework, that learns multimodal representations by maximizing the mutual information between augmented versions of these multimodal features. They prove that this framework can capture uniqueness and synergy better than existing work, and also show empirically that it leads to strong multimodal learning performance on several benchmarks.

After the discussion period, 3 reviewers voted to accept the paper, citing its strengths in theory and practice, strong results across several datasets, and broad applicability for many multimodal scenarios where capturing redundant, unique, and synergistic features are important.

The only concern blocking the paper was raised by J1jX and ZXRV regarding the feasibility of Assumption 1. In response, the authors provided empirical evidence supporting this assumption, and added detailed discussions about it in the paper. Reviewer J1jX still recommended rejection after the discussion period, but I believe that the paper's strong methodological and empirical results still warrant presentation at ICLR. Furthermore, as reviewer J1jX pointed out, I encourage the authors to significantly rephrase/relax/clarify Assumption 1, since the rest of the paper actually does not depend on it - the assumption is technically true as an existential statement, the only issue is with practically reaching such a multimodal representation using neural network approximation. I therefore lean with the 3 positive reviews and recommend acceptance.

**Additional Comments On Reviewer Discussion:**

The biggest discussion point was regarding the feasibility of Assumption 1 as raised by reviewers J1jX and ZXRV. In response, the authors provided empirical evidence supporting this assumption, and added detailed discussions about it in the paper. Reviewer J1jX still recommended rejection after the discussion period, but I believe that the paper's strong methodological and empirical results still warrant presentation at ICLR, and I encourage the authors to significantly rephrase/relax/clarify Assumption 1, since the rest of the paper actually does not depend on it - the assumption is technically true as an existential statement, the only issue is with practically reaching such a multimodal representation using neural network approximation.

There were other improvements made to the paper, including clarifications to experimental setup and a good number of new experiments added on learning synergistic features, which the sastisfied the other reviewers.

---

### Decision · Program_Chairs · 2025-01-22

Accept (Poster)